# SOFT QUALITY-DIVERSITY OPTIMIZATION

**Saeed Hedayatian[1] & Stefanos Nikolaidis[1,2]**
[1]University of Southern California
[2]Archimedes AI
{saeedhed,nikolaid}@usc.edu

## ABSTRACT

Quality-Diversity (QD) algorithms constitute a branch of optimization that is concerned with discovering a diverse and high-quality set of solutions to an optimization problem. Current QD methods commonly maintain diversity by dividing the behavior space into discrete regions, ensuring that solutions are distributed across different parts of the space. The QD problem is then solved by searching for the best solution in each region. This approach to QD optimization poses challenges in large solution spaces, where storing many solutions is impractical, and in high-dimensional behavior spaces, where discretization becomes ineffective due to the curse of dimensionality. We present an alternative framing of the QD problem, called *Soft QD*, that sidesteps the need for discretizations. We validate this formulation by demonstrating its desirable properties, such as monotonicity, and by relating its limiting behavior to the widely used QD Score metric. Furthermore, we leverage it to derive a novel differentiable QD algorithm, *Soft QD Using Approximated Diversity (SQUAD)*, and demonstrate empirically that it is competitive with current state of the art methods on standard benchmarks while offering better scalability to higher dimensional problems. Source code is available at https://github.com/conflictednerd/soft-qd

## 1 INTRODUCTION

Optimization in machine learning is typically cast as the search for a single solution that maximizes performance with respect to some objective. Quality-diversity (QD) optimization (Pugh et al., 2016; Chatzilygeroudis et al., 2021) challenges this paradigm by instead discovering a collection of solutions that are both high-performing and behaviorally diverse. This perspective is especially powerful in domains with multiple useful optima, or where robustness and user choice matter as much as raw performance. To illustrate, consider the task of painting a portrait. A traditional optimizer might aim for the single image that is most similar to the subject. Conversely, a QD optimizer not only aims for high fidelity, but also explores a *behavior space* that captures stylistic dimensions like color palette, brushstroke texture, and degree of abstraction, yielding a set of portraits that all resemble the subject while spanning a spectrum of artistic expressions. Such diversity is valuable for human selection and for escaping the limitations of optimizers (Qian et al., 2024) and imperfect objectives.

In recent years, QD optimization has grown from its roots in evolutionary computation into a broadly applicable machine learning paradigm. In reinforcement learning, QD has generated diverse policies that facilitate exploration and improve robustness, in both single (Pierrot et al., 2022; Faldor et al., 2023; Batra et al., 2024) and multi-agent (Ingvarsson et al., 2023) settings. In the context of large foundation models, QD has been adopted for red-teaming and safety analysis (Samvelyan et al., 2024; Wang et al., 2025) as well as diverse content generation (Bradley et al., 2024; Ding et al., 2024). Beyond these, QD has found applications in scenario generation (Bhatt et al., 2022; Fontaine & Nikolaidis, 2022; Zhang et al., 2024), creative design (McCormack & Gambardella, 2022; Zammit et al., 2024), engineering (Sfikas et al., 2023; Hagg et al., 2025), robotics (Huber et al., 2023; Zhong et al., 2023), and scientific discovery (Boige et al., 2023; Janmohamed et al., 2024).

QD algorithms typically operate by partitioning the behavior space into discrete cells and seeking the best solution in each cell (a tessellation, together with the stored solutions, is often referred to as the *archive*). Progress on this objective is commonly measured by the QD Score (Pugh et al., 2016), which sums the performance of the best solutions across all occupied cells, thus capturing

both quality and coverage. This approach has fueled much of the progress in QD, including recent advances that incorporate surrogate models (Gaier et al., 2018; Zhang et al., 2022), gradient information (Nilsson & Cully, 2021; Fontaine & Nikolaidis, 2021; 2023), and alternative archive structures (Vassiliades et al., 2018; Fontaine & Nikolaidis, 2023; Mouret, 2023). Yet, the formulation also presents two fundamental limitations. First, the non-differentiable nature of tessellations precludes direct optimization using gradient-based optimizers that dominate modern machine learning, except through heuristics (Fontaine & Nikolaidis, 2021; Nilsson & Cully, 2021) Second, discretizing the behavior space suffers from the curse of dimensionality in high-dimensional setting, as either the number or the volume of cells will grow exponentially, forcing methods to rely on dimensionality reduction techniques such as PCA or autoencoders (Paolo et al., 2020; Grillotti & Cully, 2021; Hedayatian & Nikolaidis, 2025). As a result, QD optimization remains hindered in high-dimensional, gradient-rich machine learning domains, despite its promise.

We address these challenges by rethinking the formulation of QD optimization itself. We introduce *Soft QD Score* as a new objective for QD optimization that measures how well a collection of solutions cover the behavior space with high-quality solutions, without discretizing the behavior space. Building on it, we derive *SQUAD*, a novel algorithm that leverages a tractable lower bound of the Soft QD score to enable end-to-end differentiable optimization. This approach has an intuitive interpretation as finding an equilibrium between attractive forces, which drive solutions toward higher quality, and repulsive forces, which spread them across the behavior space. Through experiments on both established and newly designed QD benchmarks, we demonstrate that SQUAD broadens the applicability of QD and provide further insights into its properties.

Our contributions are threefold: 1. We introduce Soft QD, a new formulation of QD optimization and analyze its theoretical properties. 2. We develop SQUAD, a differentiable QD algorithm derived from the aforementioned formulation. 3. We evaluate SQUAD on multiple benchmarks, showing its effectiveness in high-dimensional and complex optimization settings.

## 2 BACKGROUND

The quality diversity (QD) problem aims to find a collection of high-quality solutions that are diverse in their behavior. A QD problem is defined by a solution space $\Theta$, an *objective or quality function* $f : \Theta \to \mathbb{R}$ which measures a solution's quality, and a *behavior descriptor function* $\mathrm{desc} : \Theta \to \mathcal{B}$ that maps each solution to a point in the *behavior space* $\mathcal{B}$. The goal is to discover for each point in $\mathcal{B}$ a high-quality solution that exhibits that specific behavior. We can formalize this objective as finding a set of solutions $\boldsymbol{\theta} = \{\theta_b\}_{b \in \mathcal{B}}$ that maximizes the integral of their quality over the behavior space, $\int_{\mathcal{B}} f(\theta_b) \, \mathrm{d}b$. The problem is referred to as Differentiable Quality-Diversity (DQD) (Fontaine & Nikolaidis, 2021) when both the objective and descriptor functions are differentiable.

Since the behavior space $\mathcal{B}$ is continuous, QD algorithms often partition it into $n$ cells, $\mathcal{A} = \{c_1, \ldots, c_n\}$, known as an *archive* or *tessellation*. The QD objective is then framed as finding a high-quality solution for each cell. This is captured by the *QD Score*, which is the sum of the maximum quality found within each cell:

$$\max_{\boldsymbol{\theta}} \mathrm{QD\,Score}_{\mathcal{A}}(\boldsymbol{\theta}) = \sum_{c \in \mathcal{A}} \max\{f(\theta) : \theta \in \boldsymbol{\theta}, \, \mathrm{desc}(\theta) \in c\}. \tag{1}$$

Discretizing the behavior space introduces a fundamental challenge in high dimensions due to the curse of dimensionality. Grid archives (e.g., Cully et al. (2015)) divide the space evenly along each dimension, which makes the number of cells grow exponentially with the dimensionality of $\mathcal{B}$. This makes it infeasible to maintain a fine-grained grid when $\mathcal{B}$ is high-dimensional. Centroidal Voronoi Tessellation (CVT) archives (Vassiliades et al., 2018) address this by fixing the number of cells and defining them around a set of centroids. Each cell contains all points that are closer to its assigned centroid than to any other centroid, creating an almost uniform partitioning of the space. While this avoids the exponential increase of the number of cells, the volume of each CVT cell still grows exponentially with dimensionality. Large cells make it difficult to explore new regions by building on existing solutions (which is commonly done in QD methods), since reaching a different cell would likely require substantial changes to a current solution. Consequently, both discretization strategies face practical limitations in high-dimensional behavior spaces, either requiring an infeasibly large number of cells or forcing exploration across cells that are too large to navigate effectively.

## 3 SOFT QUALITY-DIVERSITY

To overcome the challenges of discretizing the behavior space, we introduce *Soft QD Score*, an objective for quality-diversity that forgoes tessellations. Conceptually, our approach builds on the view of QD algorithms as a form of "illumination" (Cully et al., 2015). We treat each solution as a light source that illuminates the behavior space, where its brightness is proportional to its quality. A set of solutions is then evaluated based on how well they illuminate the entire behavior space. This contrasts with traditional approaches, where a discrete cell is considered fully illuminated by its single best occupant. In Soft QD, solutions contribute to illuminating multiple regions, with their influence decaying smoothly as a function of distance. Figure 1 illustrates this difference.

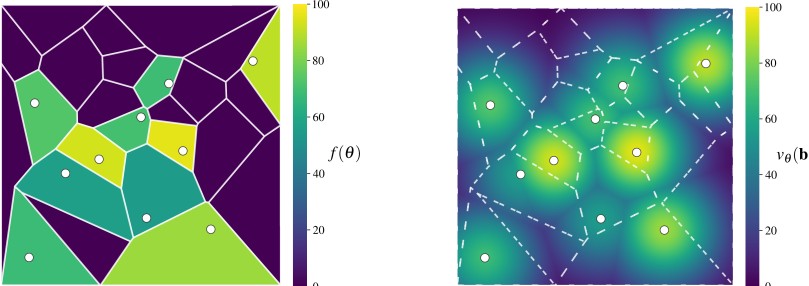

Figure 1: **Left:** In a discrete archive, each cell is fully illuminated by its highest-quality occupant. **Right:** In Soft QD, each solution illuminates the area around with an intensity proportional to its quality. The smooth scalar field defined by the behavior value $v_{\boldsymbol{\theta}}(\mathbf{b})$ is independent of discretization.

Formally, to assess a population of solutions $\boldsymbol{\theta} = \{\theta_1, \ldots, \theta_N\}$, we first define the **behavior value** $v_{\boldsymbol{\theta}}(\mathbf{b})$ that it induces at any point $\mathbf{b}$ in the behavior space as

$$v_{\boldsymbol{\theta}}(\mathbf{b}) = \max_{1 \leqslant n \leqslant N} f_n \exp\left(-\frac{\|\mathbf{b} - \mathbf{b}_n\|^2}{2\sigma^2}\right), \tag{2}$$

where $f_n = f(\theta_n)$ and $\mathbf{b}_n = \mathrm{desc}(\theta_n)$ are the quality and behavior descriptors of solution $\theta_n$, respectively, and $\sigma$ is a kernel width parameter. We use the Gaussian kernel, in line with its standard application in methods like density estimation (Bishop, 2007), to model a smooth, localized field of influence for each solution. Intuitively, $v_{\boldsymbol{\theta}}(\mathbf{b})$ measures the quality of the best available solution for a target behavior $\mathbf{b}$, discounted exponentially by its distance in behavior space. If the population contains a solution whose behavior $\mathbf{b}_n$ perfectly matches $\mathbf{b}$, the behavior value $v_{\boldsymbol{\theta}}(\mathbf{b})$ will be at least as large as its quality $f_n$. Conversely, $v_{\boldsymbol{\theta}}(\mathbf{b})$ approaches zero when there are no high-quality solutions near $\mathbf{b}$. Figure 2 illustrates how the scalar field of behavior values changes as the solutions move around in the behavior space.

The total behavior value that a population $\boldsymbol{\theta}$ induces over the entire behavior space measures its combined quality and diversity. We call this quantity **Soft QD Score** and formally define it as:

$$S(\boldsymbol{\theta}) = \int_{\mathcal{B}} v_{\boldsymbol{\theta}}(\mathbf{b}) \, \mathrm{d}\mathbf{b}. \tag{3}$$

The term "Soft" highlights a key difference from the traditional QD Score. Instead of a hard assignment of solutions to discrete cells, here each solution continuously contributes to the illumination of the behavior space. Soft QD Score captures our expectations of a QD solution. To obtain a high value, the population must contain high-quality solutions spread across the behavior space. Moreover, adding new solutions to a population will only ever increase the value of the population, and so does increasing the quality of existing solutions. The following theorem, formally stated and proven in Appendix B, establishes some of these properties. Furthermore, this theorem provides additional grounding for Soft QD Score by connecting its limiting behavior to the conventional QD Score.

**Theorem 1** (Informally stated)**.** *The Soft QD Score, as defined in Eq. 3, satisfies the following properties:*

**Monotonicity.** *The value is non-decreasing with respect to the addition of new solutions and the improvement of existing solution qualities.*
**Submodularity.** *The value is a submodular set function.*

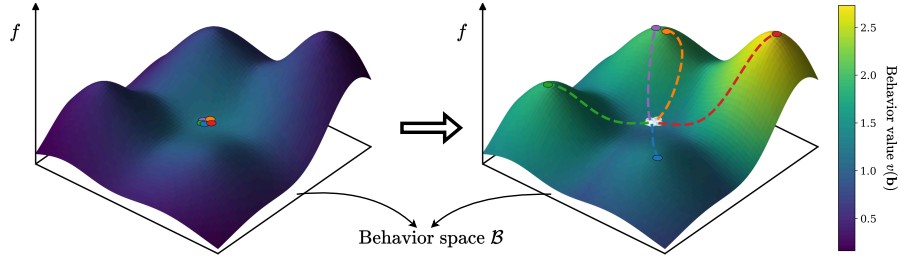

Figure 2: **Optimizing Soft QD Score with SQUAD.** The plots visualize the behavior value function, $v_{\boldsymbol{\theta}}(\mathbf{b})$, induced by a population of five solutions. The bottom plane represents the behavior space $\mathcal{B}$, the height corresponding to solution quality, $f$, and the colored surface shows the induced behavior value $v_{\boldsymbol{\theta}}$. Initially (left), a cluster of low-quality solutions induces a low behavior value. As SQUAD improves the population (right), the induced behavior value increases in both magnitude and coverage, leading to a higher Soft QD Score.

**Limiting Equivalence.** *In the limit as $\sigma \to 0$, the Soft QD Score converges (up to a constant factor) to the traditional QD score on a fine-grained archive.*

## 4 SQUAD: SOFT QD USING APPROXIMATED DIVERSITY

Directly maximizing the Soft QD Score of a population, $S(\boldsymbol{\theta})$, is challenging as it involves an integral over the behavior space. We can, however, maximize a tractable lower bound. Theorem 2 establishes such a bound, which forms the basis of our algorithm.

**Theorem 2.** *Given a population $\boldsymbol{\theta} = \{\theta_n\}_{n=1}^N$ with qualities $\{f_n\}_{n=1}^N$ and behavior descriptor vectors $\{\mathbf{b}_n\}_{n=1}^N$ in behavior space $\mathcal{B} = \mathbb{R}^d$, its Soft QD Score $S(\boldsymbol{\theta})$ can be approximated by a lower bound $\tilde{S}(\boldsymbol{\theta})$ defined as:*

$$\tilde{S}(\boldsymbol{\theta}) = (2\pi\sigma^2)^{\frac{d}{2}} \left[ \sum_{n=1}^N f_n - \sum_{1 \leqslant i < j \leqslant N} \sqrt{f_i f_j} \exp\left( -\frac{\|\mathbf{b}_i - \mathbf{b}_j\|^2}{8\sigma^2} \right) \right] \tag{4}$$

*A proof is provided in Appendix A.1.*

Our algorithm, **S**oft **Q**D **U**sing **A**pproximated **D**iversity (SQUAD), iteratively improves a randomly initialized population of $N$ solutions by updating its constituent solutions to maximize this lower bound. For brevity, we drop the leading constant $(2\pi\sigma^2)^{\frac{d}{2}}$ as it does not affect the optima. Furthermore, we rename $8\sigma^2$ as $\gamma^2$ which, as we shall see in Section 5.3, controls the quality-diversity trade-off. Therefore, with some slight abuse of notation, we define the SQUAD objective $\tilde{S}(\boldsymbol{\theta})$ as:

$$\tilde{S}(\boldsymbol{\theta}) = \sum_{n=1}^N f_n - \sum_{1 \leqslant i < j \leqslant N} \sqrt{f_i f_j} \exp\left( -\frac{\|\mathbf{b}_i - \mathbf{b}_j\|^2}{\gamma^2} \right) \tag{5}$$

Assuming that the quality and behavior descriptor functions ($f$ and $\mathrm{desc}$) are differentiable, this objective will also be fully differentiable with respect to the solutions' parameters. Hence, we can use a modern optimizer like Adam (Kingma & Ba, 2015) to iteratively update a population to improve its combined quality and diversity. This objective also has a remarkably simple interpretation. Essentially, it is composed of two summation terms:

- A **quality term** $\sum f_n$ which encourages all solutions to have higher qualities.
- A **diversity term** that acts as a pairwise repulsion, penalizing solutions that are behaviorally close.

The diversity term penalizes behavioral similarity through a sum over solution pairs. Each pair's penalty is the product of two components: the geometric mean of their qualities, $\sqrt{f_i f_j}$, and an exponential term that increases with their proximity. The combination of these two terms heavily penalizes high-quality solutions that are behaviorally similar. Notably, the geometric mean term

---

**Algorithm 1** Soft QD Using Approximated Diversity (SQUAD)

---

**Require:** Optimizer $O$, learning rate $\eta$, population size $N$, batch size $M$, neighbors $K$, epochs $T_{\max}$, kernel bandwidth $\gamma^2$.
**Require:** Differentiable evaluation function $\texttt{Eval}(\theta)$ returning quality $f$ and descriptor $\mathbf{b}$.

1: **Initialize:**
2: Population $\boldsymbol{\theta} = \{\theta_i\}_{i=1}^N$
3: Evaluations $(F, B) \leftarrow \texttt{Eval}(\boldsymbol{\theta})$
4: Optimizer state $\mathcal{S} \leftarrow O.\text{init}(\boldsymbol{\theta})$
5: **for** $t = 1$ to $T_{\max}$ **do**
6:     **for** each batch of indices $\mathcal{I} \subseteq \{1, \dots, N\}$ **do**
7:         For each $i \in \mathcal{I}$, find neighbor indices $\mathcal{N}_i \leftarrow K\text{-Nearest-Neighbors}(\mathbf{b}_i, B)$
8:         Compute objective function for batch:
$$\tilde{S}_{\mathcal{I}}(\boldsymbol{\theta}) \triangleq \sum_{i \in \mathcal{I}} f_i - \tfrac{1}{2} \sum_{i \in \mathcal{I}, j \in \mathcal{N}_i} \sqrt{f_i f_j} \exp\left(-\frac{\|\mathbf{b}_i - \mathbf{b}_j\|^2}{\gamma^2}\right)$$
9:         $G_{\mathcal{I}} \leftarrow \nabla_{\boldsymbol{\theta}_{\mathcal{I}}} \tilde{S}_{\mathcal{I}}(\boldsymbol{\theta})$
10:        Update parameters: $(\boldsymbol{\theta}_{\mathcal{I}}, \mathcal{S}_{\mathcal{I}}) \leftarrow O.\text{update}(\boldsymbol{\theta}_{\mathcal{I}}, G_{\mathcal{I}}, \mathcal{S}_{\mathcal{I}}, \eta)$
11:        Update evaluations for the batch: $(F_{\mathcal{I}}, B_{\mathcal{I}}) \leftarrow \texttt{Eval}(\boldsymbol{\theta}_{\mathcal{I}})$
12:     **end for**
13: **end for**
14: **return** Final population $\boldsymbol{\theta}$

---

discounts the similarity penalty for low-quality solutions, allowing them to first prioritize quality optimization before gradually shifting to optimize for behavioral diversity as qualities increase.

The presence of pairwise interactions in the diversity term is a direct consequence of the second-order approximation we used to derive the bound in Theorem 2. Although higher-order interactions between triplets and larger groups of solutions also exist in the full integral, our approximation considers only the most significant pairwise terms. As our analysis in Appendix A.2 shows, the component of approximation error originating from ignoring the higher-order interactions decreases as the solutions spread out. Therefore, as the solutions spread out during optimization, ignoring higher-order interactions becomes less detrimental to the accuracy of the approximation.

Building on this objective, we next describe two additional components needed for an efficient algorithm.

**Efficient computation with batches and nearest neighbors.** A naive implementation of the SQUAD objective in Eq. 5 requires computing and applying $\mathcal{O}(N^2)$ pairwise repulsions, which is computationally prohibitive for large populations. To overcome this, we only compute the repulsion for each solution from its $k$-nearest neighbors in the behavior space. This reduces the number of gradient updates that needs to be calculated at each iteration to $\mathcal{O}(Nk)$. The omission of farther solutions from the calculation is justified by the exponential decay of the repulsive force with distance, which quickly makes the contribution from distant solutions negligible. Additionally, to manage the memory and computational cost per gradient step, we update the population in mini-batches rather than all at once. In Appendix C.2 and C.1 we report the results of ablation studies on the choice of $k$ and the batch size, which show the robustness of SQUAD to these hyperparameters.

**Handling bounded behavior spaces.** Our derivation of the SQUAD objective assumes an unbounded behavior space $\mathcal{B} = \mathbb{R}^d$. However, many problems, including our experiments, have intrinsically bounded behavior descriptors. While extending the derivation to bounded domains is possible, it leads to a significantly more complex and potentially less stable final objective. We instead adopt a simpler approach: we transform the bounded space into an unbounded one using the logit function. Specifically, we map each point in the bounded behavior space $\mathbf{b} \in [0, 1]^d$ to $\mathbf{b}' = \log \frac{\mathbf{b}}{1-\mathbf{b}} \in \mathbb{R}^d$. We found this choice to be critical for the success of the algorithm, and ablated it in Appendix C.3 to confirm its importance.

Putting these pieces together, Algorithm 1 presents a complete pseudocode for SQUAD.

## 5 EXPERIMENTS

Our experiments are designed to comprehensively evaluate SQUAD and answer three key questions[1]. First, how does SQUAD scale with the dimensionality of the behavior space? Second, how does it navigate the fundamental trade-off between solution quality and population diversity? Lastly, how does SQUAD compare with state-of-the-art methods on complex, high-dimensional optimization problems? To answer these questions, we evaluate SQUAD and several baselines on three benchmark domains, each selected to probe one of these specific aspects.

### 5.1 EXPERIMENTAL SETUP

**Benchmark domains**   We evaluate different facets of QD optimization on three benchmarks, described below and in detail in Appendix D.1.
**Linear Projection (LP):** Following Fontaine et al. (2020), an algorithm must maximize an objective while maintaining diversity in a $d$-dimensional behavior space defined by a linear projection of the solution vector. We use the multi-modal Rastrigin function (Rastrigin, 1974), making this a simple yet challenging testbed for analyzing scalability with respect to $d$.
**Image Composition (IC):** Inspired by computational creativity tasks (Tian & Ha, 2022; Ibarrola & Grace, 2023), this benchmark adjusts the parameters of a set of circles (position, radius, color, transparency) to reconstruct a target image. A solution's quality is its similarity to the target image, while a 5-d behavior space encodes properties such as color harmony. The moderately sized behavior space and challenging optimization make it ideal for analyzing the quality-diversity trade-off.
**Latent Space Illumination (LSI):** Based on Fontaine & Nikolaidis (2021), algorithms search the latent space of StyleGAN2 (Karras et al., 2020) to generate images matching a target text prompt. Following Fontaine & Nikolaidis (2023), we target images of "Tom Cruise" while diversifying in age and hair length. We also propose a harder version with a 7-d behavior space in which we target images of "a detective from a noir film". Both objective and behavior descriptors use CLIP (Radford et al., 2021) embeddings to evaluate the similarity between generated images and given texts. This serves as our most difficult domain for testing QD algorithms.

**Baselines**   We compare SQUAD against state-of-the-art algorithms for high-dimensional and differentiable QD, using the open source pyribs (Tjanaka et al., 2023b) implementations: CMA-MAEGA (Fontaine & Nikolaidis, 2023), CMA-MEGA (Fontaine & Nikolaidis, 2021), and Sep-CMA-MAE (Tjanaka et al., 2023a). We also include Gradient-Assisted MAP-Elites (GA-ME), which adapts the policy-gradient-based algorithm PGA-ME Nilsson & Cully (2021) to the DQD setting by using direct gradients from the objective function (GA-ME is also similar to the OG-MAP-Elites (line) baseline in Fontaine & Nikolaidis (2021)). To ensure a fair comparison in high-dimensional behavior spaces, all baselines use Centroidal Voronoi Tesselation (CVT) to discretize the behavior space into a fixed-size archive Vassiliades et al. (2018). In addition to these methods, we also include DNS (Bahlous-Boldi et al., 2025) which is a modern variant of novelty search (Lehman & Stanley, 2011) used for QD optimization in more complex domains. We also include an improved variant of it that uses gradient-based updates (similar to GA-ME) to complement its regular mutation operator. We denote this variant as DNS-G. Additional details about the hyperparameters of the algorithms are presented in Appendix D.3.

**Evaluation metrics**   To provide a thorough analysis of the trade-offs offered by each algorithm, we use several metrics to measure the quality and diversity of the generated populations.
For quantifying diversity, we primarily use the *Vendi Score (VS)* (Friedman & Dieng, 2023), which quantifies the effective number of distinct clusters in a population. We supplement this with *Coverage* which is the percentage of occupied cells in a fixed CVT archive. To understand the quality of the generated populations, we report the *Maximum Quality* to assess pure optimization performance and the *Mean Quality* to evaluate the overall quality of solutions across the entire population. Finally, to compare the overall performances, we use *QD Score* (Pugh et al., 2016) which measures the sum of qualities in a fixed CVT archive and *Quality-weighted Vendi Score (QVS)* (Nguyen & Dieng, 2024) which extends Vendi Score to account for the quality of a population as well. A full definition of all of the metrics and further discussion is provided in Appendix D.2.

---

[1]The source code for all of our experiments is available here.

## 5.2 SCALABILITY TO HIGH-DIMENSIONAL BEHAVIOR SPACES

To evaluate SQUAD's scalability to high-dimensional behavior spaces, we conducted three sets of experiments on the LP benchmark, using 4, 8, and 16-dimensional behavior spaces. As shown in Figure 3, methods that leverage gradient information from descriptors (SQUAD, CMA-M(A)EGA) perform much better than methods that do not (Sep-CMA-MAE, GA-ME, DNS) which highlights the difficulty of exploring high-dimensional behavior spaces without gradients. To assess statistical significance of the results in Figure 3, we evaluated algorithm performance per task and metric using Kruskal-Wallis tests (all $p < 0.001$) followed by Holm-Bonferroni-corrected Mann-Whitney U tests to compare the best algorithm against the others. All differences were significant ($p < 0.001$) except for CMA-MAEGA on medium ($d = 8$) and hard ($d = 16$) tasks for QD Scores, where it was not significantly different from the top-performing algorithms; in the hard domain, this is primarily due to the high variance of CMA-MAEGA.

While CMA-MEGA and CMA-MAEGA have a slight edge in the 4-dimensional behavior space, SQUAD closes this gap and noticeably outperforms them in the more challenging versions of the task. We attribute the initial success of CMA-M(A)EGA to their large archives ($10^4$ cells), which are dense enough to effectively cover the low-dimensional space. However, as the dimensionality increases, the density of their archives drops exponentially, making the feedback less informative. This limitation is a key reason for their performance decline in higher-dimensional spaces. SQUAD, on the other hand, does not discretize the behavior space which enables it to maintain strong performance as dimensionality increases. Furthermore, SQUAD demonstrates greater stability across all three tasks, with the lowest variance across different evaluations. More detailed results of these experiments are presented in Appendix E.

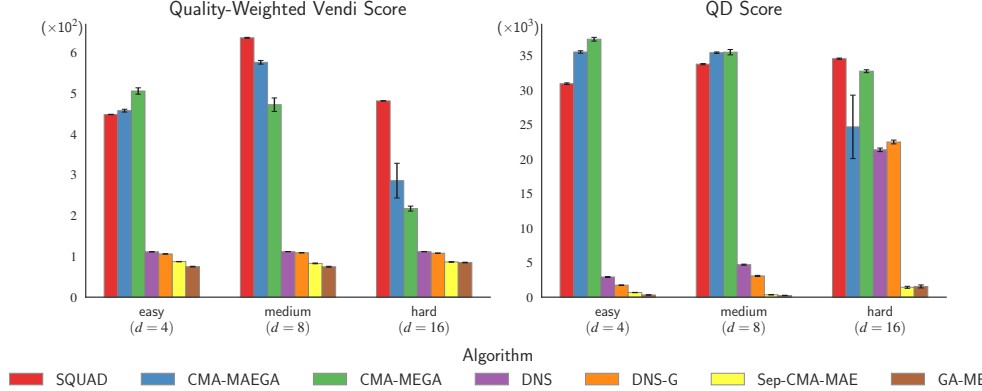

Figure 3: **QVS (left) and QD Score (right) on LP tasks with increasing behavior descriptor dimensionality (4, 8, 16).** All results are averaged over 10 runs, with error bars depicting the standard errors. SQUAD's performance relative to the baselines improves with task complexity, with it outperforming all other methods on the most challenging 16-d task for both metrics.

## 5.3 ANALYSIS OF THE QUALITY-DIVERSITY TRADE-OFFS

Obtaining diversity often comes at the expense of quality. In this section, we shed some light on the quality-diversity tradeoff that SQUAD offers in comparison to baseline algorithms using the IC domain, which provides a realistic optimization challenge with a moderately sized, 5-d behavior space. We also examined the bandwidth parameter, $\gamma^2$, and showed that it acts a as a knob, allowing SQUAD to effectively trade off quality for diversity.

As the results in Table 1 indicate, SQUAD outperforms the baselines on most metrics. The average quality of the solutions found by SQUAD is significantly higher than those of the baselines. Furthermore, the high quality of the best solution in SQUAD's population shows that it is highly capable in pure quality optimization. On diversity metrics, SQUAD maintains its noticeable lead on Vendi Score but falls short of CMA-MAEGA on Coverage by a small margin. The discrepancy between the two diversity metrics can be explained by the fact that Vendi Score takes the shape of the archive into account. Therefore, while the larger number of solutions found by CMA-MAEGA helps it

Table 1: **Performance in the IC domain.** Comparing SQUAD ($\gamma^2 = 1$) with baselines in terms of the quality (Best Objective, Mean Objective) and diversity (Vendi Score, Coverage). Results are mean $\pm$ standard error averaged over 10 runs, with the best score for each metric shown in **bold**.

| Algorithm | Quality | | Diversity | |
|---|---|---|---|---|
| | Mean Objective | Max Objective | Vendi Score | Coverage |
| SQUAD | $\mathbf{83.37 \pm 0.02}$ | $\mathbf{93.58 \pm 0.10}$ | $\mathbf{5.49 \pm 0.00}$ | $5.68 \pm 0.06$ |
| CMA-MAEGA | $74.83 \pm 0.20$ | $88.79 \pm 0.92$ | $3.93 \pm 0.03$ | $\mathbf{5.85 \pm 0.05}$ |
| CMA-MEGA | $75.98 \pm 0.26$ | $86.18 \pm 1.58$ | $3.25 \pm 0.24$ | $4.54 \pm 0.49$ |
| DNS | $71.30 \pm 0.15$ | $74.54 \pm 0.18$ | $1.62 \pm 0.01$ | $1.52 \pm 0.02$ |
| DNS-G | $74.49 \pm 0.03$ | $76.48 \pm 0.11$ | $1.67 \pm 0.00$ | $1.49 \pm 0.03$ |
| Sep-CMA-MAE | $72.15 \pm 0.21$ | $74.57 \pm 0.04$ | $1.32 \pm 0.01$ | $0.46 \pm 0.02$ |
| GA-ME | $73.44 \pm 0.41$ | $74.53 \pm 0.45$ | $1.14 \pm 0.04$ | $0.19 \pm 0.03$ |

cover the behavior space more densely, this coverage is concentrated in a smaller region, leading to its lower Vendi Score. This is further supported by the archive visualizations in Appendix E.

Having compared SQUAD with the baselines, we examined how to control its quality-diversity tradeoff via the bandwidth parameter, $\gamma^2$. As Equation 5 suggests, increasing $\gamma^2$ boosts the contribution of the diversity term by increasing its effective range and intensity. Empirically, we confirmed this by evaluating SQUAD with different values of $\gamma^2$ ranging from $10^{-3}$ to 50 in the IC domain. As Figure 4 shows, increasing $\gamma^2$ indeed improves the diversity of the solutions, albeit at the price of their quality. These results show how SQUAD lets the users trade off quality for diversity (and vice versa) by changing the value of a single hyperparameter.

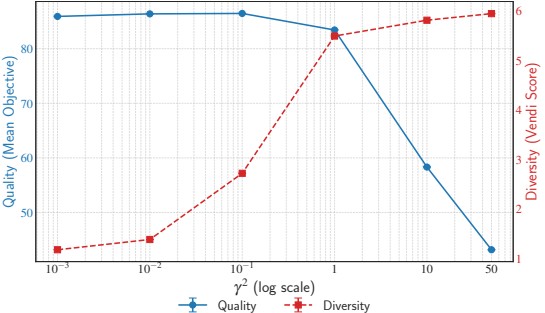

Figure 4: **Controlling the quality-diversity trade-off with $\gamma^2$.** The plot shows how varying $\gamma^2$ impacts solution quality, measured by the mean objective (blue line), and diversity, measured by the Vendi Score (dashed red line).

### 5.4 PERFORMANCE ON CHALLENGING DQD PROBLEMS

In our last set of experiments, we compared SQUAD with baselines on the challenging LSI domain. In line with prior work (Fontaine & Nikolaidis, 2023) we searched the latent space of StyleGAN2 for latents that would generate images of Tom Cruise and are diverse with respect to age and hair length. Additionally, we also set up a more challenging version of the LSI benchmark with a 7-d behavior space. In this more difficult task the goal is to generate "photos of a detective from a noir film" and the images have to be diverse with respect to different attributes including facial expression, pose, and hair color. Additional details about both versions of LSI is presented in Appendix D.1.3.

Table 2: **Performance in the Latent Space Illumination (LSI) domain.** Results are averaged over 5 runs and reported as mean $\pm$ standard error, with the best scores highlighted in **bold**. Algorithms that failed to achieve a positive mean objective (hence, have a zero QVS) are shown with $^*$.

| Algorithm | LSI | | LSI (Hard) | |
|---|---|---|---|---|
| | QD Score $_{(\times 10^3)}$ | QVS | QD Score $_{(\times 10^3)}$ | QVS |
| SQUAD | $\mathbf{13.41 \pm 0.19}$ | $\mathbf{177.0 \pm 2.8}$ | $\mathbf{2.55 \pm 0.08}$ | $\mathbf{151.3 \pm 0.1}$ |
| CMA-MAEGA | $6.82 \pm 0.10$ | $121.6 \pm 9.7$ | $0.39 \pm 0.07$ | $99.3 \pm 1.0$ |
| CMA-MEGA | $8.70 \pm 0.07$ | $140.1 \pm 1.7$ | $0.27 \pm 0.05$ | $92.8 \pm 1.5$ |
| DNS | $-9.31 \pm 2.52$ | $0.0 \pm 0.0^*$ | $-11.27 \pm 1.46$ | $0.0 \pm 0.0^*$ |
| DNS-G | $-8.53 \pm 1.50$ | $0.0 \pm 0.0^*$ | $-6.81 \pm 0.47$ | $0.0 \pm 0.0^*$ |
| Sep-CMA-MAE | $-0.59 \pm 0.45$ | $0.0 \pm 0.0^*$ | $0.02 \pm 0.04$ | $0.0 \pm 0.0^*$ |
| GA-ME | $-14.90 \pm 1.67$ | $0.0 \pm 0.0^*$ | $0.08 \pm 0.00$ | $0.0 \pm 0.0^*$ |

The performance of different algorithms are compared in terms of QD Score and Quality-weighted Vendi Score in Table 2. In both LSI tasks, SQUAD outperforms the baselines by a significant margin, showcasing its capability in challenging domains. A closer inspection of the data (available in Appendix E) reveals that SQUAD is particularly adept in exploring the behavior space. Thus, despite performing similar to the baselines in terms of mean quality, SQUAD's ability to maintain a better coverage of the behavior space differentiates it from the baselines. Similar to our prior results, Sep-CMA-MAE and GA-ME perform poorly on both versions of the LSI task. Here, we attribute their poor performance to the high-dimensional and non-linear nature of the optimization landscape, which makes it difficult to navigate the behavior space. Interestingly, GA-ME, despite using gradient ascent updates, is unable to effectively escape local optima, highlighting the important role of modern optimizers in more challenging domains.

## 6 RELATED WORK

Quality-diversity (QD) optimization has recently become a topic of interest in machine learning, following the introduction of the NSLC (Lehman & Stanley, 2011) and MAP-Elites (Cully et al., 2015) algorithms. MAP-Elites, as a canonical QD algorithm, uses random mutations to find solutions that occupy different behavioral *niches* in a tessellated behavior space. This general recipe is improved by methods that have proposed better discretization schemes (Vassiliades et al., 2018; Mouret, 2023), incorporated modern evolutionary optimizers (Fontaine et al., 2020; Tjanaka et al., 2023a; Batra et al., 2024; Choi & Togelius, 2021), and improved the search by using gradient-aware mutations (Nilsson & Cully, 2021; Pierrot et al., 2022; Faldor et al., 2023) and crossovers (Ingvarsson et al., 2023). It is worth noting that these advancements rely on discrete archives, which are known to be sensitive to the archive resolution (Fontaine & Nikolaidis, 2023). To the best of our knowledge, the only other work that proposes a discretization-free formulation of QD is the continuous QD Score of Kent et al. (2022). Continuous QD Score is similar to SoftQD Score in that both compute an integral over the behavior space. However, continuous QD Score employs a non-smooth kernel, which makes it difficult to approximate analytically and restricts estimation to Monte Carlo sampling methods. Consequently, it is used exclusively as an evaluation metric rather than as an optimization objective. Another major contribution was the formulation of differentiable QD (Fontaine & Nikolaidis, 2021) where the quality and behavior descriptor functions are assumed to be differentiable. This, along with the introduction of gradient arborescence algorithms (Fontaine & Nikolaidis, 2023) paved the way for QD algorithms to tackle large scale optimization problems (Yu et al., 2025; Wan et al., 2025). Our work is a continuation of this trend that frames the whole QD problem as a unified optimization problem, enabling seamless integration with modern gradient-based optimizers.

The pairwise repulsive term in SQUAD's objective is reminiscent of the kernel-based repulsion used in particle variational inference methods such as Stein Variational Gradient Descent (SVGD) (Liu & Wang, 2016), where particles are simultaneously attracted toward regions of high target density and repelled from one another to prevent collapse. Unlike QD, these methods aim to approximate a probability distribution rather than optimize quality while diversifying in a behavior space, which does not exist in SVGD or its reinforcement learning variant, SVPG (Liu et al., 2017). Nonetheless, recent SVGD advances for high-dimensional inference, including message passing (Zhuo et al., 2018), matrix-valued kernels (Wang et al., 2019), and Newton-like updates (Chen et al., 2019), offer practical ideas for stabilizing and scaling repulsive forces that could be adapted to SQUAD's objective. Lastly, a related example outside variational inference is DOMiNO (Zahavy et al., 2023), which enforces diversity in reinforcement learning via repulsive forces and a Lagrangian formulation, showing the broader utility of such mechanisms for balancing quality with diversity.

## 7 CONCLUSION

In this work, we introduced Soft QD as a new formulation of QD optimization that does not require discretizing the behavior space. Building on it, we proposed SQUAD, a novel QD algorithm tailored for differentiable domains. Our experiments across multiple benchmarks demonstrated that SQUAD achieves competitive performance with state-of-the-art methods, and exhibits promising scalability to higher-dimensional behavior spaces. In addition, we highlighted how it enables a convenient trade-off between quality and diversity, offering a fresh perspective on the design of QD algorithms.

Despite the encouraging results, several limitations remain. First, the current formulation of SQUAD assumes differentiable objectives and behavior descriptors, which may be costly to obtain or unavailable in certain domains. Extending Soft QD to reinforcement learning settings with estimated gradients, or to fully non-differentiable domains via evolutionary strategies, represents a promising direction, where issues such as navigating deceptive behavior landscapes may pose unique challenges. Second, our analysis in Section 5.3 showed the critical role of the kernel bandwidth, $\gamma$ in shaping the quality-diversity trade-off of SQUAD. Future work could investigate adaptive or per-solution schedules for $\gamma$, for example by adjusting it based on the distribution of solutions or by annealing it during training. Moreover, while SQUAD's second-order approximation of the Soft QD Score offers tractability, it also discards higher-order interactions. Alternatives such as sparsification of the interactions (Spielman & Srivastava, 2011), message-passing (Zhuo et al., 2018), or Monte Carlo approaches may provide richer modeling of interactions, though potentially at higher computational cost. Lastly, while we use a logit transformation to handle bounded behavior spaces, exploring alternative transformations could be an avenue for future work to further improve performance. Taken together, we introduce Soft QD as a powerful conceptual tool and SQUAD as a practical algorithm, highlighting a new path for quality-diversity in complex, differentiable domains. We hope this work inspires the community to build upon this foundation, realizing the full promise of scalable and general purpose QD.

## 8 ACKNOWLEDGMENTS AND DISCLOSURE OF FUNDING

We would like to thank Varun Bhatt, Sophie Hsu, Bryon Tjanaka, and Shihan Zhao for their feedback on a preliminary version of this work. This work has been partially supported by the NSF CAREER #2145077, NSF NRI #2024949 and the DARPA EMHAT project.

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

# A  LOWER BOUNDING SOFT QD SCORE

## A.1  APPROXIMATE SOFT QD SCORE

Here, we will provide a formal proof of Theorem 2.

Similar to the main paper, let $\Theta$ be a parameter space and $\boldsymbol{\theta} = \{\theta_1, \ldots, \theta_N\}$ be a set of $N$ solutions where $\theta_n \in \Theta$. Furthermore, let $f : \Theta \to [0, \infty)$ be an objective function that assigns higher values to better solutions, and let $\mathrm{desc} : \Theta \to \mathbb{R}^d$ be the behavior descriptor function that quantifies the behavior of each solution with a $d$-dimensional vector. Throughout the proofs, we use $\mathbf{b}_i$ and $f_i$ as shorthands for $\mathrm{desc}(\theta_i)$ and $f(\theta_i)$, respectively. The Soft QD Score of the population $\boldsymbol{\theta}$ is defined as

$$S(\boldsymbol{\theta}) = \int_{\mathcal{B}} v_{\boldsymbol{\theta}}(\mathbf{b}) \, \mathrm{d}\mathbf{b} = \int_{\mathcal{B}} \left[ \max_{1 \leqslant n \leqslant N} f_n \exp\left( -\frac{\|\mathbf{b} - \mathbf{b}_n\|^2}{2\sigma^2} \right) \right] \mathrm{d}\mathbf{b}. \tag{6}$$

**Theorem 2.** *Given a population $\boldsymbol{\theta} = \{\theta_n\}_{n=1}^N$ with qualities $\{f_n\}_{n=1}^N$ and behavior descriptor vectors $\{\mathbf{b}_n\}_{n=1}^N$ in behavior space $\mathcal{B} = \mathbb{R}^d$, its Soft QD Score $S(\boldsymbol{\theta})$ can be approximated by a lower bound $\tilde{S}(\boldsymbol{\theta})$ defined as:*

$$\tilde{S}(\boldsymbol{\theta}) = (2\pi\sigma^2)^{\frac{d}{2}} \left[ \sum_{n=1}^N f_n - \sum_{1 \leqslant i < j \leqslant N} \sqrt{f_i f_j} \exp\left( -\frac{\|\mathbf{b}_i - \mathbf{b}_j\|^2}{8\sigma^2} \right) \right] \tag{7}$$

*Proof.* To make the notation simpler, let us define the *contribution of a single solution* $\theta_n$ *at a point* $\mathbf{b} \in \mathcal{B}$ as

$$g_n(\mathbf{b}) = f_n \exp\left( -\frac{\|\mathbf{b} - \mathbf{b}_n\|^2}{2\sigma^2} \right). \tag{8}$$

The behavior value $v_{\boldsymbol{\theta}}(\mathbf{b})$ is the maximum of these individual contributions:

$$v_{\boldsymbol{\theta}}(\mathbf{b}) = \max_n g_n(\mathbf{b}) \tag{9}$$

Using Maximum-minimums identity, we can rewrite this as

$$v_{\boldsymbol{\theta}}(\mathbf{b}) = \max_n g_n(\mathbf{b}) \tag{10}$$

$$= \sum_i g_i(\mathbf{b}) - \sum_{i<j} \min\left( g_i(\mathbf{b}), g_j(\mathbf{b}) \right) + \sum_{i<j<k} \min\left( g_i(\mathbf{b}), g_j(\mathbf{b}), g_k(\mathbf{b}) \right) - \ldots \tag{11}$$

Therefore,

$$S(\boldsymbol{\theta}) = \int \max_n g_n(\mathbf{b}) \, \mathrm{d}\mathbf{b} \tag{12}$$

$$= \sum_i \int g_i(\mathbf{b}) \, \mathrm{d}\mathbf{b} \tag{13}$$

$$- \sum_{i<j} \int \min\left( g_i(\mathbf{b}), g_j(\mathbf{b}) \right) \mathrm{d}\mathbf{b} \tag{14}$$

$$+ \sum_{i<j<k} \int \min\left( g_i(\mathbf{b}), g_j(\mathbf{b}), g_k(\mathbf{b}) \right) \mathrm{d}\mathbf{b} \tag{15}$$

$$- \ldots \tag{16}$$

To get a tractable lower bound on this, we truncate the series and only consider the first two sums. This discards the higher-order interactions involving three or more solutions. Intuitively, if the solutions are well-spread in the behavior space (i.e., $\|\mathbf{b}_i - \mathbf{b}_j\|^2 \gg 2\sigma^2$) this approximation is acceptable (detailed error analysis is provided in the next section). The second order-approximation, $\tilde{S}(\boldsymbol{\theta})$, is therefore obtained by only keeping the individual and pairwise effects:

$$S(\boldsymbol{\theta}) \approx \tilde{S}(\boldsymbol{\theta}) = \sum_i \int g_i(\mathbf{b}) \, \mathrm{d}\mathbf{b} - \sum_{i<j} \int \min\left( g_i(\mathbf{b}), g_j(\mathbf{b}) \right) \mathrm{d}\mathbf{b} \tag{17}$$

Next, we will derive a closed-form solution for each of the two terms.

**The Quality Term (Individual Contributions):** The integrals in the first sum are standard Gaussian integrals and represent the contribution of each solution to the Score, irrespective of other solutions. We can evaluate them analytically:

$$\int_{\mathbb{R}^d} g_i(\mathbf{b})\, \mathrm{d}\mathbf{b} = \int_{\mathbb{R}^d} f_i \exp\left(-\frac{\|\mathbf{b} - \mathbf{b}_i\|^2}{2\sigma^2}\right)\, \mathrm{d}\mathbf{b} = f_i(2\pi\sigma^2)^{\frac{d}{2}} \tag{18}$$

Hence, the first sum in $\tilde{S}(\boldsymbol{\theta})$ evaluates to

$$\sum_{i=1}^{N} \int_{\mathbb{R}^d} g_i(\mathbf{b})\, \mathrm{d}\mathbf{b} = (2\pi\sigma^2)^{\frac{d}{2}} \sum_{i=1}^{N} f_i. \tag{19}$$

**The Diversity Term (Pairwise Overlaps):** The integrals in the second term are more difficult to compute and require yet another approximation. Note that for any pair of non-negative numbers $x, y$ we have $\min(x, y) \leqslant \sqrt{xy}$ (geometric mean). Using this, we have the following upper bound approximation:

$$\int_{\mathbb{R}^d} \min(g_i(\mathbf{b}), g_j(\mathbf{b}))\, \mathrm{d}\mathbf{b} \approx \int_{\mathbb{R}^d} \sqrt{g_i(\mathbf{b})g_j(\mathbf{b})}\, \mathrm{d}\mathbf{b}. \tag{20}$$

Now note that

$$\sqrt{g_i(\mathbf{b})g_j(\mathbf{b})} = \sqrt{f_i \exp\left(-\frac{\|\mathbf{b} - \mathbf{b}_i\|^2}{2\sigma^2}\right) f_j \exp\left(-\frac{\|\mathbf{b} - \mathbf{b}_j\|^2}{2\sigma^2}\right)} \tag{21}$$

$$= \sqrt{f_i f_j}\, \exp\left(-\frac{\|\mathbf{b} - \mathbf{b}_i\|^2 + \|\mathbf{b} - \mathbf{b}_j\|^2}{4\sigma^2}\right). \tag{22}$$

The exponent is a quadratic in $\mathbf{b}$ and can be simplified as

$$\|\mathbf{b} - \mathbf{b}_i\|^2 + \|\mathbf{b} - \mathbf{b}_j\|^2 = \|\mathbf{b}\|^2 - 2\mathbf{b} \cdot \mathbf{b}_i + \|\mathbf{b}_i\|^2 + \|\mathbf{b}\|^2 - 2\mathbf{b} \cdot \mathbf{b}_j + \|\mathbf{b}_j\|^2 \tag{23}$$

$$= 2\|\mathbf{b}\|^2 - 2\mathbf{b} \cdot (\mathbf{b}_i + \mathbf{b}_j) + \|\mathbf{b}_i\|^2 + \|\mathbf{b}_j\|^2 \tag{24}$$

$$= 2\left\|\mathbf{b} - \frac{\mathbf{b}_i + \mathbf{b}_j}{2}\right\|^2 - 2\left\|\frac{\mathbf{b}_i + \mathbf{b}_j}{2}\right\|^2 + \|\mathbf{b}_i\|^2 + \|\mathbf{b}_j\|^2 \tag{25}$$

$$= 2\left\|\mathbf{b} - \frac{\mathbf{b}_i + \mathbf{b}_j}{2}\right\|^2 + \frac{1}{2}\left(-\|\mathbf{b}_i\|^2 - 2\mathbf{b}_i \cdot \mathbf{b}_j - \|\mathbf{b}_j\|^2 + 2\|\mathbf{b}_i\|^2 + 2\|\mathbf{b}_j\|^2\right) \tag{26}$$

$$= 2\left\|\mathbf{b} - \frac{\mathbf{b}_i + \mathbf{b}_j}{2}\right\|^2 + \frac{1}{2}(\|\mathbf{b}_i\|^2 - 2\mathbf{b}_i \cdot \mathbf{b}_j + \|\mathbf{b}_j\|^2) \tag{27}$$

$$= 2\left\|\mathbf{b} - \frac{\mathbf{b}_i + \mathbf{b}_j}{2}\right\|^2 + \frac{1}{2}\|\mathbf{b}_i - \mathbf{b}_j\|^2. \tag{28}$$

Substituting this back into the exponential, we get

$$\exp\left(-\frac{1}{4\sigma^2}\left[2\left\|\mathbf{b} - \frac{\mathbf{b}_i + \mathbf{b}_j}{2}\right\|^2 + \frac{1}{2}\|\mathbf{b}_i - \mathbf{b}_j\|^2\right]\right) = \exp\left(-\frac{\|\mathbf{b}_i - \mathbf{b}_j\|^2}{8\sigma^2}\right) \exp\left(-\frac{\|\mathbf{b} - \frac{\mathbf{b}_i + \mathbf{b}_j}{2}\|^2}{2\sigma^2}\right). \tag{29}$$

The overlap integral is therefore

$$\int_{\mathbb{R}^d} \sqrt{g_i g_j}\, \mathrm{d}\mathbf{b} = \sqrt{f_i f_j} \exp\left(-\frac{\|\mathbf{b}_i - \mathbf{b}_j\|^2}{8\sigma^2}\right) \int_{\mathbb{R}^d} \exp\left(-\frac{\|\mathbf{b} - \frac{\mathbf{b}_i + \mathbf{b}_j}{2}\|^2}{2\sigma^2}\right)\, \mathrm{d}\mathbf{b}, \tag{30}$$

where the integral is now just a regular Gaussian integral centered at $\frac{\mathbf{b}_i + \mathbf{b}_j}{2}$. The whole integral thus evaluates to

$$\sqrt{f_i f_j}(2\pi\sigma^2)^{d/2} \exp\left(-\frac{\|\mathbf{b}_i - \mathbf{b}_j\|^2}{8\sigma^2}\right) \tag{31}$$

Putting these all together, we establish a closed form approximation (lower bound) of the archive score:

$$\tilde{S}(\boldsymbol{\theta}) = (2\pi\sigma^2)^{\frac{d}{2}} \left[ \sum_{n=1}^{N} f_n - \sum_{1 \leqslant i < j \leqslant N} \sqrt{f_i f_j} \exp\left( -\frac{\|\mathbf{b}_i - \mathbf{b}_j\|^2}{8\sigma^2} \right) \right] \tag{32}$$

$\square$

### A.2 ANALYSIS OF APPROXIMATION ERROR

The error between the true Soft QD Score of a population and the second order approximation derived above stems from two sources: (1) the truncation of higher-order interaction in the maximum-minimums equality (Equation 17), and (2) the replacing of pairwise minimums with their geometric means in the integrals (Equation 20). We will analyze each of these errors and discuss how we can control them.

**Truncation Error:** The truncation error for the second order approximation is

$$\varepsilon_1 = |S(\boldsymbol{\theta}) - \tilde{S}(\boldsymbol{\theta})| \tag{33}$$

$$= \left| \int \max_n g_n(\mathbf{b}) \, d\mathbf{b} - \left[ \sum_i \int g_i(\mathbf{b}) \, d\mathbf{b} - \sum_{i<j} \int \min\left(g_i(\mathbf{b}), g_j(\mathbf{b})\right) \, d\mathbf{b} \right] \right| \tag{34}$$

$$= \left| \sum_{i<j<k} \int \min\left(g_i(\mathbf{b}), g_j(\mathbf{b}), g_k(\mathbf{b})\right) \, d\mathbf{b} - \cdots \right| \tag{35}$$

**Lemma 1** (Bonferroni Inequalities for the Maximum). *Let $\{x_1, \ldots, x_N\}$ be a set of non-negative real numbers and let $S_m = \sum_{|I|=m} \min_{i \in I} x_i$. The partial sums of the maximum-minimums equality, $P_K = \sum_{m=1}^{K} (-1)^{m-1} S_m$, provide alternating bounds on the maximum value:*

- *If K is odd, $P_K \geqslant \max(x_1, \ldots, x_N)$.*
- *If K is even, $P_K \leqslant \max(x_1, \ldots, x_N)$.*

*Proof.* For bounding the approximation error, we only need to use this lemma with $K = 2, 3$, which we shall prove. Without loss of generality, assume that the numbers are sorted such that $x_1 \geqslant x_2 \geqslant \cdots \geqslant x_N \geqslant 0$. The contribution of $x_k$ to the sum $S_m$ is $\binom{k-1}{m-1} x_k$, since $x_k$ is the minimum of a subset of $m$ variables iff the other $m-1$ elements are all chosen from the $k-1$ elements smaller than it.

The partial sums $P_K$ can thus be written as

$$P_K = \sum_{m=1}^{K} (-1)^{m-1} S_m = \sum_{k=1}^{N} \left[ \sum_{m=1}^{K} (-1)^{m-1} \binom{k-1}{m-1} \right] x_k. \tag{36}$$

The inner sum is a partial sum of a binomial expansion. Therefore, using the fact that

$$\sum_{j=0}^{m} (-1)^j \binom{n}{j} = (-1)^m \binom{n-1}{m}, \tag{37}$$

(which follows from induction on $m$) we can see that the coefficient of $x_k$ in $P_K$ is

$$C(k, K) = \sum_{j=0}^{\min(k-1,K-1)} (-1)^j \binom{k-1}{j} = (-1)^{\min(k-1,K-1)} \binom{k-2}{\min(k-1, K-1)}. \tag{38}$$

Let us now consider the cases of $K = 2$ and $K = 3$ individually.

**Case $K = 2$:** Consider $P_2 = \sum_i x_i - \sum_{i<j} \min(x_i, x_j)$. For all $k \geqslant 2$, the coefficient of $x_k$ in $P_2$ is $2 - k$ and $x_1$ is only present once, with coefficient 1 in $S_1$. Therefore, we have

$$P_2 = x_1 + \sum_{k=2}^{N}(2 - k)x_k. \tag{39}$$

Since $x_k$'s are all non-negative and $2 - k$ is non-positive, the sum is non-positive. Therefore,

$$\boxed{P_2 \leqslant x_1 = \max(x_1, \ldots, x_N).} \tag{40}$$

**Case $K = 3$:** Consider $P_3 = P_2 + \sum_{i<j<k} \min(x_i, x_j, x_k)$. For all $k \geqslant 3$, the coefficient of $x_k$ in $P_3$ is $\binom{k-2}{2} = \frac{(k-2)(k-3)}{2}$. Furthermore, $x_1$ is only present once, with coefficient 1 in $S_1$ and $x_2$ is present only twice, once with a $+1$ coefficient in $S_1$ and once with coefficient $-1$ in $S_2$. Therefore, we have

$$P_3 = x_1 + 0 \cdot x_2 + \sum_{k=3}^{N} \frac{(k-2)(k-3)}{2}x_k. \tag{41}$$

Since $x_k$'s are all non-negative and so are all the coefficients, the sum is non-negative. Therefore,

$$\boxed{P_3 \geqslant x_1 = \max(x_1, \ldots, x_N)} \tag{42}$$

$\square$

**Lemma 2** (Bounding the Truncation Error). *The truncation error $\varepsilon_1$ is bounded by the sum of the integrals of the third-order minimums:*

$$\varepsilon_1 \leqslant \sum_{i<j<k} \int \min(g_i(\mathbf{b}), g_j(\mathbf{b}), g_k(\mathbf{b})) \, d\mathbf{b} \tag{43}$$

*Proof.* The Bonferroni inequality for $K = 2$ states that for any point $\mathbf{b}$

$$\max_n g_n(\mathbf{b}) \geqslant \sum_i g_i(\mathbf{b}) - \sum_{i<j} \min(g_i(\mathbf{b}), g_j(\mathbf{b})). \tag{44}$$

Integrating this pointwise inequality over the domain of $\mathbf{b}$ yields:

$$S(\boldsymbol{\theta}) = \int \max_k g_n(\mathbf{b}) \, d\mathbf{b} \geqslant \int \left[\sum_i g_i(\mathbf{b}) - \sum_{i<j} \min(g_i(\mathbf{b}), g_j(\mathbf{b}))\right] d\mathbf{b} = \tilde{S}(\boldsymbol{\theta}) \tag{45}$$

Therefore, $\tilde{S}(\boldsymbol{\theta})$ is indeed a lower bound on $S(\boldsymbol{\theta})$ and we can write the error as $\varepsilon_1 = S(\boldsymbol{\theta}) - \tilde{S}(\boldsymbol{\theta})$. Similarly, the Bonferroni inequality for $K = 3$ states that for any point $\mathbf{b}$

$$\max_n g_n(\mathbf{b}) \leqslant \sum_i g_i(\mathbf{b}) - \sum_{i<j} \min(g_i(\mathbf{b}), g_j(\mathbf{b})) + \sum_{i<j<k} \min(g_i(\mathbf{b}), g_j(\mathbf{b}), g_k(\mathbf{b})). \tag{46}$$

Integrating over the domain of $\mathbf{b}$ yields:

$$S(\boldsymbol{\theta}) = \int \max_n g_n(\mathbf{b}) \, d\mathbf{b} \tag{47}$$

$$\leqslant \int \left[\sum_i g_i(\mathbf{b}) - \sum_{i<j} \min(g_i(\mathbf{b}), g_j(\mathbf{b})) + \sum_{i<j<k} \min(g_i(\mathbf{b}), g_j(\mathbf{b}), g_k(\mathbf{b}))\right] d\mathbf{b} \tag{48}$$

$$= \tilde{S}(\boldsymbol{\theta}) + \int \sum_{i<j<k} \min(g_i(\mathbf{b}), g_j(\mathbf{b}), g_k(\mathbf{b})) \, d\mathbf{b} \tag{49}$$

$$\implies S(\boldsymbol{\theta}) - \tilde{S}(\boldsymbol{\theta}) \leqslant \int \sum_{i<j<k} \min(g_i(\mathbf{b}), g_j(\mathbf{b}), g_k(\mathbf{b})) \, d\mathbf{b} \tag{50}$$

$$\implies \varepsilon_1 \leqslant \sum_{i<j<k} \int \min(g_i(\mathbf{b}), g_j(\mathbf{b}), g_k(\mathbf{b})) \, d\mathbf{b} \tag{51}$$

$\square$

Given the above, we use the same technique that we used before and replace minimum with geometric mean to compute the integral. We can see that

$$\int \min(g_i(\mathbf{b}), g_j(\mathbf{b}), g_k(\mathbf{b})) \, d\mathbf{b} \leqslant \int (g_i(\mathbf{b}) \cdot g_j(\mathbf{b}) \cdot g_k(\mathbf{b}))^{\frac{1}{3}} \, d\mathbf{b} \tag{52}$$

$$= (f_i f_j f_k)^{\frac{1}{3}} \int \exp\left(-\frac{\|\mathbf{b} - \mathbf{b}_i\|^2 + \|\mathbf{b} - \mathbf{b}_j\|^2 + \|\mathbf{b} - \mathbf{b}_k\|^2}{6\sigma^2}\right) d\mathbf{b} \tag{53}$$

The integral is the product of three Gaussians. By completing the square, similar to the pairwise case, we can rewrite it as a Gaussian centered around $\frac{\mathbf{b}_i + \mathbf{b}_j + \mathbf{b}_k}{3}$ and a constant term. The result would be

$$\int (g_i(\mathbf{b}) \cdot g_j(\mathbf{b}) \cdot g_k(\mathbf{b}))^{\frac{1}{3}} \, d\mathbf{b} = (f_i f_j f_k)^{\frac{1}{3}} (3\pi\sigma^2)^{\frac{d}{2}} \exp\left(-\frac{\|\mathbf{b}_i - \mathbf{b}_j\|^2 + \|\mathbf{b}_j - \mathbf{b}_k\|^2 + \|\mathbf{b}_i - \mathbf{b}_k\|^2}{18\sigma^2}\right) \tag{54}$$

So, we have

$$\varepsilon_1 \leqslant \sum_{i<j<k} (f_i f_j f_k)^{\frac{1}{3}} \left(\frac{2\pi\sigma^2}{3}\right)^{\frac{d}{2}} \exp\left(-\frac{\|\mathbf{b}_i - \mathbf{b}_j\|^2 + \|\mathbf{b}_j - \mathbf{b}_k\|^2 + \|\mathbf{b}_i - \mathbf{b}_k\|^2}{18\sigma^2}\right) \tag{55}$$

From this, we conclude that if the solutions are well separated (i.e., the mean squared distance of any triplet is significantly larger than $6\sigma^2$) the error becomes negligible .

**Pairwise Integral Approximation Error** The pairwise integral approximation error is due to the estimation of pairwise minimums using geometric means. For every pair of solutions $(i, j)$ this error is

$$\epsilon_{i,j} = \int_{\mathcal{B}} \left(\sqrt{g_i(\mathbf{b})g_j(\mathbf{b})} - \min\left(g_i(\mathbf{b}), g_j(\mathbf{b})\right)\right) d\mathbf{b}. \tag{56}$$

It follows from the arithmetic-geometric inequality that for every non-negative numbers $x$ and $y$, $\sqrt{xy} - \min(x,y) \leqslant \frac{1}{2}|x - y|$. Integrating over $\mathbf{b}$ preserves this inequality, yielding

$$\epsilon_{i,j} \leqslant \frac{1}{2} \int_{\mathcal{B}} |g_i(\mathbf{b}) - g_j(\mathbf{b})| \, d\mathbf{b}. \tag{57}$$

Note that the right hand side is just the $\ell_1$ distance between the functions $g_i, g_j$. This distance can be further simplified by breaking down $|g_i - g_j|$ using the triangle inequality. Without loss of generality, assume that $f_j \leqslant f_i$, then add and subtract $f_j \exp\left(-\frac{\|\mathbf{b} - \mathbf{b}_i\|^2}{2\sigma^2}\right)$, we get

$$|g_i(\mathbf{b}) - g_j(\mathbf{b})| \leqslant |f_i - f_j| \exp\left(-\frac{\|\mathbf{b} - \mathbf{b}_i\|^2}{2\sigma^2}\right) + f_j \left|\exp\left(-\frac{\|\mathbf{b} - \mathbf{b}_i\|^2}{2\sigma^2}\right) - \exp\left(-\frac{\|\mathbf{b} - \mathbf{b}_j\|^2}{2\sigma^2}\right)\right| \tag{58}$$

Now, if we integrate both sides, the first term in the right hand side is just a multiple of a Gaussian integral. So we would get

$$\int |g_i(\mathbf{b}) - g_j(\mathbf{b})| \, d\mathbf{b} \leqslant (2\pi\sigma^2)^{\frac{d}{2}} \left(|f_i - f_j| + f_j \int_{\mathcal{B}} |p_i(\mathbf{b}) - p_j(\mathbf{b})| \, d\mathbf{b}\right), \tag{59}$$

where $p_i(\mathbf{b})$ is the pdf of a Gaussian centered at $\mathbf{b}_i$ with covariance $\sigma^2 \mathbf{I}$. The remaining integral is twice the Total Variation distance $d_{TV}(p_i, p_j)$. Using Pinsker's inequality, we can transform it into the KL divergence between two Gaussians, which does have a closed form solution, $\frac{\|\mathbf{b}_i - \mathbf{b}_j\|^2}{2\sigma^2}$. Replacing the integral and combining everything, we arrive at the following bound for the error:

$$\epsilon_{i,j} \leqslant (2\pi\sigma^2)^{\frac{d}{2}} \left(|f_i - f_j| + \min(f_i, f_j)\frac{\|\mathbf{b}_i - \mathbf{b}_j\|}{\sigma}\right) \tag{60}$$

Finally, using the triangle inequality, we see that

$$\varepsilon_2 \leqslant (2\pi\sigma^2)^{\frac{d}{2}} \sum_{i<j\leqslant K} \left(|f_i - f_j| + \min(f_i, f_j)\frac{\|\mathbf{b}_i - \mathbf{b}_j\|}{\sigma}\right). \tag{61}$$

Putting both of these bounds together, a final application of triangle inequality shows that

$$|S(\boldsymbol{\theta}) - \tilde{S}(\boldsymbol{\theta})| \leqslant \varepsilon_1 + \varepsilon_2, \tag{62}$$

where $\varepsilon_1$ and $\varepsilon_2$ are themselves bounded by Equation 55 and Equation 61, respectively.

# B    PROPERTIES OF SOFT QD SCORE

Here, we will further discuss some properties of the Soft QD Score that make it a suitable measure of quality and diversity. Theorems 3, 4, 5, and 6 formalize and prove the statement of Theorem 1 in the main paper and show multiple properties of the Soft QD Score.

Theorems 3 and 4 show that SoftQD Score is monotonic with respect to population size and qualities of members of the population. This means that adding new solutions to a population and improving the quality of existing ones will never decrease the SoftQD Score. We note that this is a desirable, yet non-trivial property of a measure of quality and diversity. For instance, neither mean objective value (as a measure of quality) nor the mean behavior distance (as a measure of diversity) of a population are monotonic with respect to the population size; that is, adding a new solution can decrease them.

Theorem 5 shows that SoftQD Score is also submodular. Being submodular is particularly convenient as it implies the existence of efficient approximate algorithms for maximizing it under certain conditions. For instance, it is well known that maximizing a submodular function subject to a cardinality constraint admits a $1 - \frac{1}{e}$ approximation algorithm (Nemhauser et al., 1978). This is valuable when we want to select a fixed-size subset of a large population that preserves as much of the Soft QD Score as possible (e.g., for evaluation or compression).

Lastly, we establish a connection between the traditional QD Score and the limiting behavior of the Soft QD Score through Theorem 6.

**Theorem 3** (Monotonicity with respect to population size). *Let $\boldsymbol{\theta} = \{\theta_1, \ldots, \theta_N\}$ be a population and let $\theta_{N+1}$ be any new solution. If $\boldsymbol{\theta}^+ = \boldsymbol{\theta} \cup \{\theta_{N+1}\}$, then $S(\boldsymbol{\theta}) \leqslant S(\boldsymbol{\theta}^+)$.*

*Proof.* Let $f_{N+1} = f(\theta_{N+1})$ and $\mathbf{b}_{N+1} = \mathrm{desc}(\theta_{N+1})$. The behavior value for the new population $\boldsymbol{\theta}^+$ can be written as

$$v_{\boldsymbol{\theta}^+}(\mathbf{b}) = \max\left(v_{\boldsymbol{\theta}}(\mathbf{b}), f_{N+1} \exp\left(-\frac{\|\mathbf{b} - \mathbf{b}_{N+1}\|^2}{2\sigma^2}\right)\right) \geq v_{\boldsymbol{\theta}}(\mathbf{b}). \tag{63}$$

Integrating both sides of the above inequality over the behavior space $\mathcal{B}$ yields the result:

$$S(\boldsymbol{\theta}^+) = \int_{\mathcal{B}} v_{\boldsymbol{\theta}^+}(\mathbf{b})\, \mathrm{d}\mathbf{b} \geqslant \int_{\mathcal{B}} v_{\boldsymbol{\theta}}(\mathbf{b})\, \mathrm{d}\mathbf{b} = S(\boldsymbol{\theta}). \tag{64}$$

$\square$

**Theorem 4** (Monotonicity with respect to quality). *Let $\boldsymbol{\theta} = \{\theta_1, \ldots, \theta_N\}$ be a population and $\boldsymbol{\theta}' = \boldsymbol{\theta} \cup \{\theta'_n\}\backslash\{\theta_n\}$ be another population that is identical to $\boldsymbol{\theta}$ except that the $n$-th solution $\theta_n$ is replaced by $\theta'_n$ such that $\mathrm{desc}(\theta) = \mathrm{desc}(\theta'_n) = \mathbf{b}_n$ and $f(\theta'_n) = f'_n \geqslant f_n = f(\theta_n)$. Then, $S(\boldsymbol{\theta}) \leqslant S(\boldsymbol{\theta}')$.*

*Proof.* Let the behavior values for $\boldsymbol{\theta}$ and $\boldsymbol{\theta}'$ be $v_{\boldsymbol{\theta}}(\mathbf{b})$ and $v_{\boldsymbol{\theta}'}(\mathbf{b})$, respectively. Since, $f'_n \geqslant f_n$, for every $\mathbf{b}$ we have $f'_n \exp(-\frac{\|\mathbf{b} - \mathbf{b}_n\|^2}{2\sigma^2}) \geqslant f_n \exp(-\frac{\|\mathbf{b} - \mathbf{b}_n\|^2}{2\sigma^2})$. Therefore, we can write

$$v_{\boldsymbol{\theta}'}(\mathbf{b}) = \max\left(v_{\boldsymbol{\theta}}(\mathbf{b}), f'_n \exp\left(-\frac{\|\mathbf{b} - \mathbf{b}_n\|^2}{2\sigma^2}\right)\right) \geq v_{\boldsymbol{\theta}}(\mathbf{b}). \tag{65}$$

Integrating both sides of the above inequality yields the result:

$$S(\boldsymbol{\theta}') = \int_{\mathcal{B}} v_{\boldsymbol{\theta}'}(\mathbf{b})\, \mathrm{d}\mathbf{b} \geqslant \int_{\mathcal{B}} v_{\boldsymbol{\theta}}(\mathbf{b})\, \mathrm{d}\mathbf{b} = S(\boldsymbol{\theta}). \tag{66}$$

$\square$

**Theorem 5** (Submodularity). *Let $\boldsymbol{\theta} = \{\theta_1, \ldots, \theta_N\}$ be a population. The Soft QD Score $S$ defined on subsets of $\boldsymbol{\theta}$ is submodular. That is, for any $U \subseteq V \subseteq \boldsymbol{\theta}$ and any new solution $\theta' \notin V$ with quality $f' = f(\theta')$ and behavior vector $\mathbf{b}' = \mathrm{desc}(\theta')$,*

$$S(U \cup \{\theta'\}) - S(U) \geqslant S(V \cup \{\theta'\}) - S(V). \tag{67}$$

*Proof.* For brevity, let us denote $U \cup \{\theta'\}$ as $U'$ and $V \cup \{\theta'\}$ as $V'$. Similar to the argument in Theorem 3, we have

$$v_{U'}(\mathbf{b}) - v_U(\mathbf{b}) = \max\left(v_U(\mathbf{b}), f' \exp\left(-\frac{\|\mathbf{b} - \mathbf{b}'\|^2}{2\sigma^2}\right)\right) - v_U(\mathbf{b}) \tag{68}$$

$$= \max\left(0, f' \exp\left(-\frac{\|\mathbf{b} - \mathbf{b}'\|^2}{2\sigma^2}\right) - v_U(\mathbf{b})\right). \tag{69}$$

Similarly, we also have

$$v_{V'}(\mathbf{b}) - v_V(\mathbf{b}) = \max\left(0, f' \exp\left(-\frac{\|\mathbf{b} - \mathbf{b}'\|^2}{2\sigma^2}\right) - v_V(\mathbf{b})\right). \tag{70}$$

Since $U \subseteq V$, for every $\mathbf{b}$ we have $v_U(\mathbf{b}) \leqslant v_V(\mathbf{b})$. Therefore,

$$v_{U'}(\mathbf{b}) - v_U(\mathbf{b}) \geqslant v_{V'}(\mathbf{b}) - v_V(\mathbf{b}) \tag{71}$$

Integrating both sides yields the result

$$S(U') - S(U) \geqslant S(V') - S(V). \tag{72}$$

$\square$

**Theorem 6.** *Let $\boldsymbol{\theta} = \{\theta_1, \ldots, \theta_N\}$ be a population of $N$ solutions with corresponding quality values $f_1, \ldots, f_N$ and distinct behaviors $\mathbf{b}_1, \ldots, \mathbf{b}_N$ in $\mathbb{R}^d$. Let $S(\boldsymbol{\theta})$ be the Soft QD Score defines as*

$$S(\boldsymbol{\theta}) = \int_{\mathbb{R}^d} \max_{1 \leqslant n \leqslant N}\left[f_n \exp\left(-\frac{\|\mathbf{b} - \mathbf{b}_n\|^2}{2\sigma^2}\right)\right] d\mathbf{b} \tag{73}$$

*In the limit as the kernel width $\sigma$ approaches zero, the scaled Soft QD Score converges to the sum of the qualities of all solutions in the population.*

$$\lim_{\sigma \to 0} \frac{S(\boldsymbol{\theta})}{(2\pi\sigma^2)^{\frac{d}{2}}} = \sum_{n=1}^{N} f_n \tag{74}$$

*This limit is equivalent to the traditional QD Score calculated on a grid fine enough to isolate each solution into its own cell.*

*Proof.* Let the scaled Soft QD Score be denoted by $L(\sigma)$:

$$L(\sigma) = \frac{S(\boldsymbol{\theta})}{(2\pi\sigma^2)^{\frac{d}{2}}} = \frac{1}{(2\pi\sigma^2)^{\frac{d}{2}}} \int_{\mathbb{R}^d} \max_{1 \leqslant n \leqslant N}\left[f_n \exp\left(-\frac{\|\mathbf{b} - \mathbf{b}_n\|^2}{2\sigma^2}\right)\right] d\mathbf{b} \tag{75}$$

Since the sum of a set of non-negative number is at least as large as their maximum, we can replace the max in the integral with a summation and write

$$L(\sigma) \leqslant \frac{1}{(2\pi\sigma^2)^{\frac{d}{2}}} \int_{\mathbb{R}^d} \sum_{n=1}^{N}\left[f_n \exp\left(-\frac{\|\mathbf{b} - \mathbf{b}_n\|^2}{2\sigma^2}\right)\right] d\mathbf{b} \tag{76}$$

$$= \frac{1}{(2\pi\sigma^2)^{\frac{d}{2}}} \sum_{n=1}^{N} f_n \int_{\mathbb{R}^d} \exp\left(-\frac{\|\mathbf{b} - \mathbf{b}_n\|^2}{2\sigma^2}\right) d\mathbf{b} \tag{77}$$

$$= \sum_{n=1}^{N} f_n. \tag{78}$$

Note that Eq. 77 holds due to the linearity of integration and Eq. 78 is true since the Gaussian integrals over the entire domain evaluate to $(2\pi\sigma^2)^{\frac{d}{2}}$.

Next, let $r = \frac{1}{2} \min_{i \neq j} \|\mathbf{b}_i - \mathbf{b}_j\|$. Following this definition, we can see that the open balls $B_n = \{\mathbf{b} : \|\mathbf{b} - \mathbf{b}_n\| < r\}$ centered at each behavior point with radius $r$ are disjoint.

The integrand in $S(\boldsymbol{\theta})$ is non-negative, so the integral over $\mathbb{R}^d$ is greater than or equal to the integral over the union of these disjoint balls:

$$S(\boldsymbol{\theta}) \geqslant \int_{\cup_n B_n} \max_i \left[ f_i \exp\left( -\frac{\|\mathbf{b} - \mathbf{b}_i\|^2}{2\sigma^2} \right) \right] d\mathbf{b} \tag{79}$$

$$= \sum_n \int_{B_n} \max_i \left[ f_i \exp\left( -\frac{\|\mathbf{b} - \mathbf{b}_i\|^2}{2\sigma^2} \right) \right] d\mathbf{b} \tag{80}$$

For any point $\mathbf{b}$ inside a specific ball $B_n$, the maximum value is always greater than or equal to the term for $n$:

$$\max_i \left[ f_i \exp\left( -\frac{\|\mathbf{b} - \mathbf{b}_i\|^2}{2\sigma^2} \right) \right] \geqslant f_n \exp\left( -\frac{\|\mathbf{b} - \mathbf{b}_n\|^2}{2\sigma^2} \right) \tag{81}$$

Substituting this, gives a lower bound for $S(\boldsymbol{\theta})$

$$S(\boldsymbol{\theta}) \geqslant \sum_n f_n \int_{B_n} \exp\left( -\frac{\|\mathbf{b} - \mathbf{b}_n\|^2}{2\sigma^2} \right) d\mathbf{b} \tag{82}$$

Now, we can analyze the limit of the scaled version of this lower bound:

$$\lim_{\sigma \to 0} L(\sigma) \geqslant \lim_{\sigma \to 0} \sum_n \frac{f_n}{(2\pi\sigma^2)^{\frac{d}{2}}} \int_{B_n} \exp\left( -\frac{\|\mathbf{b} - \mathbf{b}_n\|^2}{2\sigma^2} \right) d\mathbf{b} \tag{83}$$

$$= \sum_n f_n \left( \lim_{\sigma \to 0} \frac{1}{(2\pi\sigma^2)^{\frac{d}{2}}} \int_{B_n} \exp\left( -\frac{\|\mathbf{b} - \mathbf{b}_n\|^2}{2\sigma^2} \right) d\mathbf{b} \right) \tag{84}$$

Similar to the integral of Eq. 77, The term in the parenthesis here is the integral of the PDF of $\mathcal{N}(\mathbf{b}_n, \sigma^2 I)$. Since the ball $B_n$ is a fixed region containing the mean $\mathbf{b}_n$, all of the probability mass concentrates inside $B_n$ as $\sigma \to 0$. Therefore, the limit of the integral approaches the denominator and the whole limit will evaluate to one and we have

$$\lim_{\sigma \to 0} L(\sigma) \geqslant \sum_n f_n \tag{85}$$

Together, Eq. 85 and Eq. 76 sandwich $L(\sigma)$. Hence,

$$\lim_{\sigma \to 0} L(\sigma) = \sum_{n=1}^N f_n. \tag{86}$$

Lastly, note that if the solutions in $\boldsymbol{\theta}$ all fall into distinct cells of an archive, as is the case with the populations that conventional QD methods generate or in the limit of having very fine-grained archives, the sum of the objectives coincides with the QD Score and we have

$$\lim_{\sigma \to 0} \frac{S(\boldsymbol{\theta})}{(2\pi\sigma^2)^{\frac{d}{2}}} = \sum_{n=1}^N f_n = \text{QD Score}(\boldsymbol{\theta}) \tag{87}$$

$\square$

## C  ADDITIONAL ABLATIONS

### C.1  EFFECT OF BATCH SIZE

SQUAD updates the solutions in its population one batch at a time (Section 4), primarily to reduce the computational cost of simultaneous updates. To examine the role of batch size $M$, we performed an ablation in the IC domain, evaluating SQUAD with $M \in \{4, 8, 16, 32, 64\}$. Each setting was repeated three times with different random seeds. The results are reported in Table 3.

Overall, the performance of SQUAD is stable across all tested batch sizes. Smaller batch sizes tend to yield slightly higher QD scores, whereas larger batch sizes achieve marginally better QVS. However, these differences are minor compared to the variation observed between algorithms, indicating that SQUAD is robust to the choice of batch size. All experiments were conducted on a machine equipped with an NVIDIA GeForce RTX 4090. The approximate runtimes for different batch sizes were as follows: batch size $4$ took $5$ hours and $47$ minutes, batch size $8$ took $2$ hours and $55$ minutes, batch size $16$ took $2$ hours and $42$ minutes, batch size $32$ took $3$ hours and $10$ minutes, batch size $64$ took $3$ hours and $10$ minutes.

Table 3: **Effect of batch size on SQUAD.** Results report the mean and standard error over three random seeds for five batch sizes. Experiment in the main paper use $M = 64$.

| Algorithm | QD Score ($\times 10^3$) | QVS |
|---|---|---|
| SQUAD ($M = 4$) | **5.32 $\pm$ 0.05** | 457.4 $\pm$ 0.4 |
| SQUAD ($M = 8$) | 5.12 $\pm$ 0.04 | 457.7 $\pm$ 0.4 |
| SQUAD ($M = 16$) | 5.04 $\pm$ 0.16 | **458.4 $\pm$ 0.5** |
| SQUAD ($M = 32$) | 4.99 $\pm$ 0.09 | 457.9 $\pm$ 0.3 |
| SQUAD ($M = 64$) | 5.07 $\pm$ 0.05 | 457.4 $\pm$ 0.1 |

## C.2 Effect of Number of Neighbors

We ablate the sensitivity of SQUAD to the number of nearest neighbors $k$ used in the diversity term of the objective. Experiments were run in the IC domain with $k \in \{0, 4, 8, 16, 32\}$ and the results over three random seeds are reported in Table 4. Note that the case $k = 0$ removes the diversity term entirely and therefore corresponds to independently optimizing each solution for quality alone.

Table 4: **Effect of nearest neighbors' count on SQUAD.** Results report the mean and standard error over three random seeds. Experiment in the main paper use $k = 16$.

| Algorithm | QD Score ($\times 10^3$) | QVS |
|---|---|---|
| SQUAD ($k = 0$) | 0.09 $\pm$ 0.00 | 90.8 $\pm$ 0.0 |
| SQUAD ($k = 4$) | **5.37 $\pm$ 0.06** | 457.7 $\pm$ 0.4 |
| SQUAD ($k = 8$) | 5.02 $\pm$ 0.03 | **459.7 $\pm$ 0.5** |
| SQUAD ($k = 16$) | 5.07 $\pm$ 0.05 | 457.4 $\pm$ 0.1 |
| SQUAD ($k = 32$) | 5.19 $\pm$ 0.04 | 448.5 $\pm$ 0.6 |

Two conclusions follow from these results. First, the diversity term is essential as the $k = 0$ baseline achieves significantly worse performance in both QD Score and QVS, suggesting a collapse into a set of nearly identical solutions. Second, for other values of $k > 0$, the performance is largely insensitive to the exact number of neighbors, since neither QD Score nor QVS vary largely across $k = 4, 8, 16, 32$. In practice, this means that a small $k$ already provides the repulsive pressure needed to encourage diverse and high quality solutions.

Our hypothesis for this observed insensitivity is that once solutions are separated by distances larger than the kernel bandwidth $\gamma$, which could happen early in the optimization, the kernel contribution from farther solutions rapidly decays towards zero, so only the closest neighbors exert meaningful repulsion. This hypothesis, combined with our results indicating the important role that $\gamma$ plays, also suggests several promising directions for future work: (1) per-solution or adaptive kernel widths $\gamma$ based on the current spread of solutions in the behavior space to control the repulsive force each solution receives; possibly combined with (2) annealing schedules that dampen the diversity term over training so that exploration is encouraged early on and fine-grained quality optimization dominates later.

## C.3 Effect of Behavior Space Transformation

Recall that the derivation of SQUAD's objective (Equation 5) assumed that the behavior space is unbounded. To transform the bounded behavior space $[0, 1]^d$ that is used in the tasks to $\mathbb{R}^d$, SQUAD used the logit transformation:

$$\mathbf{b}' = \log \frac{\mathbf{b}}{1 - \mathbf{b}}, \tag{88}$$

where all operations are performed element-wise. In our final ablation, we seek to understand the impact of this transformation on SQUAD's performance. To this end, we run SQUAD without using this transformation with three different random seeds and compare the results with the original experiments that were conducted in Section 5.3 (conducted over ten random seeds). As the comparison of the results in Table 5 shows, the logit transformation is critical to the successful performance of SQUAD.

Table 5: **Effect of behavior space transformation on SQUAD.** Results report the mean and standard error over three random seeds for the ablated version and over ten random seeds for the base version.

| Algorithm | QD Score ($\times 10^3$) | QVS |
|---|---|---|
| SQUAD | $5.09 \pm 0.05$ | $457.3 \pm 0.2$ |
| SQUAD (w/o behavior space transformation) | $2.96 \pm 0.04$ | $186.6 \pm 0.8$ |

### C.4 EFFECT OF POPULATION SIZE

We also study the impact that role that population size has on the performance of SQUAD. To this end, we compare the performance of SQUAD in the IC domain with 4 different population sizes $N = 128, 256, 512, 1024$. The results, reported in Table 6, conform with our expectation that larger populations yield better performance. Furthermore, we observed that the runtime of SQUAD had a linear relation with the population size, with the mean runtimes being 24, 48, 95, 190 minutes for population sizes of 128, 256, 512, and 1024, respectively (averaged over 3 seeds, with variances less than 10 seconds).

Table 6: **Effect of population size on SQUAD.** Results report the mean and standard error over three random seeds. Experiment in the main paper use $N = 1024$.

| Algorithm | QD Score ($\times 10^3$) | QVS |
|---|---|---|
| SQUAD ($N = 128$) | $3.24 \pm 0.43$ | $390.6 \pm 1.6$ |
| SQUAD ($N = 256$) | $4.12 \pm 0.57$ | $422.7 \pm 0.8$ |
| SQUAD ($N = 512$) | $4.26 \pm 0.60$ | $445.2 \pm 0.3$ |
| SQUAD ($N = 1024$) | $\mathbf{4.50 \pm 0.62}$ | $\mathbf{457.5 \pm 0.6}$ |

## D IMPLEMENTATION DETAILS

### D.1 BENCHMARK DOMAINS

#### D.1.1 LINEAR PROJECTION

We adopt the Linear Projection (LP) domain introduced by Fontaine et al. (2020), using the Rastrigin objective function (Rastrigin, 1974). The QD search is performed over solution vectors $\mathbf{x} \in \mathbb{R}^n$, with dimensionality set to $n = 1024$. The Rastrigin function is defined as

$$f_{\text{Rastrigin}}(\mathbf{x}) = 10n + \sum_{i=1}^{n} [x_i^2 - 10\cos(2\pi x_i)]. \tag{89}$$

Following prior work (Fontaine et al., 2019), we restrict the search space to $[-5.12, 5.12]^n$ and apply an offset so that the global optimum is shifted from the origin to $[\underbrace{2.048, \ldots, 2.048}_{n}]^T$. To transform the problem from minimization to maximization, we normalize the objective via a linear transformation

$$f(\mathbf{x}) = 100 \times \frac{M - f_{Rastrigin}(\mathbf{x})}{M} \tag{90}$$

where $M$ denotes the maximum value of the Rastrigin function in the search domain. This results in objective values scaled to the range $[0, 100]$. A heatmap of the transformed Rastrigin function in 2 dimensions is depicted in Figure 5.

The behavior space is defined by partitioning $\mathbf{x} = [x_1, \ldots, x_n]^T$ into $d$ equal-sized chunks and computing the mean of clipped values within each chunk. More formally, the $k$-dimensional behavior descriptor is given by

$$\text{desc}(\mathbf{x}) = \frac{1}{d} \left( \sum_{i=1}^{\frac{n}{d}} \text{clip}(x_i), \sum_{i=\frac{n}{d}+1}^{\frac{2n}{d}} \text{clip}(x_i), \ldots, \sum_{i=\frac{(d-1)n}{d}+1}^{n} \text{clip}(x_i) \right)^T, \tag{91}$$

where the clipping function is defined as

$$
\text{clip}(x_i) = \begin{cases} x_i & \text{if } -5.12 \leq x_i \leq 5.12 \\ \frac{5.12}{x_i} & \text{otherwise} \end{cases} \tag{92}
$$

In our experiments, we evaluated behavior spaces of dimensionality $d \in \{4, 8, 16\}$. For additional details on this domain, including its challenges in high-dimensional settings, we refer the reader to Fontaine & Nikolaidis (2021; 2023).

### D.1.2 IMAGE COMPOSITION



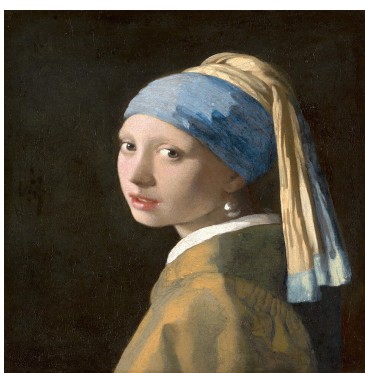

(a) **(Transformed)** 2-**d Rastrigin function**  (b) **Target image for the IC domain**

Figure 5

Image Composition (IC) is a new differentiable QD benchmark we introduce, inspired by related works (Tian & Ha, 2022; Ibarrola & Grace, 2023). The IC task aims to reconstruct a target image by composing a large number of simple primitives (circles) on a canvas. The objective is to match the target as closely as possible while exploring diverse visual effects, which are captured through five behavioral descriptors. In this task, a solution is a vector of size $n \times 7$, where $n$ denotes the number of circles (here $n = 1024$). Each row parameterizes a circle with 7 values: the $(x, y)$ center coordinates, radius, RGB color values, and an opacity coefficient. All parameters are represented as unconstrained logits, which are passed through sigmoidal transformations and rescaled to the appropriate range.

A differentiable renderer composites these circles in sequence onto a canvas of resolution $64 \times 64$. Each circle is drawn with smooth edges, controlled by a softness hyperparameter (here set to $10.0$). Rendering is performed by alpha-compositing onto a black canvas.

The objective is defined as the similarity between the rendered image and a fixed target image. We structural similarity (SSIM) (Wang et al., 2004), normalized to the range $[0, 100]$ to measure this. The behavioral descriptors are five statistics computed from the circles:

- mean radius of circles,

- variance of radii,

- variety of RGB values in the palette (color spread),

- coherence of circle hues in HSV space (color harmony),

- degree of spatial clustering based on average 5-nearest-neighbor distances.

These descriptors are each normalized to lie in $[0, 1]$, with higher values representing larger radii, greater diversity, more harmony, or tighter clustering, respectively. Together with the objective, they define a continuous and differentiable QD landscape. As the target image in our experiments, we use Johannes Vermeer's painting Girl with a Pearl Earring (Figure 5), obtained from the freely available reproduction on Wikimedia Commons.

### D.1.3 Latent Space Illumination

Latent Space Illumination (LSI) (Fontaine et al., 2021; Fontaine & Nikolaidis, 2021) is a challenging QD benchmark designed to illuminate the latent space of a generative model by discovering diverse and high-quality solutions. Following the experimental setup of Fontaine & Nikolaidis (2023); Tjanaka et al. (2023b), we employ StyleGAN2 Karras et al. (2020) as the generative model and use CLIP (Radford et al., 2021) to define both the objective and behavior descriptor functions.

Each solution in LSI is represented as a 9216-dimensional vector corresponding to a point in the latent space of StyleGAN2. To evaluate solution quality, we pass the vector through StyleGAN2 to generate an image, which is then compared to a target text prompt using CLIP embeddings. Behavior descriptors are similarly computed by comparing the generated image against pairs of descriptor sentences, one positive and one negative. Our implementation is based on JAX (Bradbury et al., 2018), and we rely on publicly available JAX-based implementations of both StyleGAN2 and CLIP.

We define two task variants:

- **Base version.** This setup follows Fontaine & Nikolaidis (2023) and uses the prompt "A photo of Tom Cruise" as the objective. Behavior descriptors are specified by two sentence pairs:
    - ("Photo of Tom Cruise as a small child", "Photo of Tom Cruise as an elderly person")
    - ("Photo of Tom Cruise with long hair", "Photo of Tom Cruise with short hair")
- **Hard version.** To increase task difficulty, we use the objective prompt "A photo of a detective from a noir film" and define seven behavior descriptor pairs:
    - ("Photo of a young kid", "Photo of an elderly person")
    - ("Photo of a person with long hair", "Photo of a person with short hair")
    - ("Photo of a person with dark hair", "Photo of a person with white hair"
    - ("Photo of a person smiling", "Photo of a person frowning")
    - ("Photo of a person with a round face", "Photo of a person with an oval face")
    - ("Photo of a person with thin, sparse hair", "Photo of a person with thick, full hair")
    - ("Photo of a person looking directly into the camera", "Photo of a person looking sideways")

### D.2 Evaluation Metrics

We used multiple evaluation metrics in our experiments, which we shall explain here in more detail.

**Vendi Score** (Friedman & Dieng, 2023) is a widely applicable metric of diversity in machine learning. Given a set of samples and a pairwise similarity function, Vendi Score can be interpreted as the effective number of unique elements in the set. Formally, it is defined as

$$\text{VS}(\mathbf{K}) = \exp\left(-\text{tr}(\frac{1}{n}\mathbf{K}\log\frac{1}{n}\mathbf{K})\right) = \exp\left(-\sum_{i=1}^{n}\lambda_i\log\lambda_i\right), \tag{93}$$

where $\mathbf{K} \in \mathbb{R}^{n\times n}$ is a positive semi-definite similarity matrix and $\lambda_i$ are the eigenvalues of $\frac{1}{n}\mathbf{K}$. In our case, the similarity matrix is derived by applying the Gaussian kernel on the distance between the behavior descriptors of solutions. That is, for solutions $i$ and $j$ with behavior descriptors $\mathbf{b}_i, \mathbf{b}_j$ we define: $\mathbf{K}_{ij} = \exp\left(-\frac{\|\mathbf{b}_i - \mathbf{b}_j\|^2}{\sigma_v^2}\right)$ where $\sigma_v$ is a kernel bandwidth. For a $d$ dimensional behavior space, we choose $\sigma_v^2 = \frac{d}{6}$ in our evaluations. This (heuristic) choice is motivated by the fact that the mean squared distance between two uniformly selected vectors in $[0, 1]^d$ is $\frac{d}{6}$. Since in all of our experiments the behavior space is defined as $[0, 1]^d$, this ensures that the similarity between two random vectors in the behavior space will be $e^{-1} \approx 0.37$.

**Quality-weighted Vendi Score (QVS)** (Nguyen & Dieng, 2024) extends Vendi Score to also incorporate the quality of solutions. It is computed by multiplying the Vendi Score by the mean quality of the solutions:

$$\text{QVS}(\mathbf{K}, (f_1, \ldots, f_n)) = \frac{1}{n}\sum_{i=1}^{n} f_i \, \text{VS}(\mathbf{K}), \tag{94}$$

where $f_i$ is the quality of the $i$-th solutions. We use QVS to capture the joint effect of quality and diversity in a population. Importantly, the mean quality of the solutions must be non-negative for the metric to be meaningful. While in our domains the objectives are normalized such that sensible solutions have objectives in the $[0, 100]$ range, it is still possible for out-of-bound solutions in the LP and LSI domains to obtain negative values, since, theoretically, their objective functions are unbounded from below. In cases where the mean objective of the solutions were negative (which only happened with Sep-CMA-MAE and GA-ME in LSI), we report the QVS as $0.0$ and report the fine-grained statistics, including the mean objective and Vendi Score, in Appendix E.

We also leverage a discretization of the behavior space using CVT (Vassiliades et al., 2018) and report traditional QD metrics such as **QD Score** (Pugh et al., 2016) and **Coverage**. Even though the exponentially growing volume of the cells hinders the performance of optimization algorithms that leverage such archives (as noted in the paper), we can still use them for evaluation. To compute these metrics, we discretize the behavior space into a fixed number of cells ($1024$ for IC and $512$ for LP and LSI) and insert the solutions from a population into the resulting archive, keeping only the best solution that lands in a cell. The **Coverage** is then defined as the fraction of cells that are filled with solutions and captures the diversity of the population. In a same manner **QD Score** is defined as the sum of the qualities (objectives) of all the solutions in the resulting archive, and captures both quality and diversity.

Lastly, we also use **Mean Objective** and **Max Objective** to measure the quality of populations. These are simply defined as the mean (and respectively maximum) of the qualities (objectives) of the solutions in a population.

## D.3    HYPERPARAMETERS

The full set of hyperparameters that we used for the experiments can be found in the accompanying code. Here, we will go over the most important choices.

### D.3.1    BASELINES

All baselines use a CVT archive with $10^4$ cells in LP and IC domains and a finer archive with $4 \times 10^4$ cells in the LSI domain. In LP and IC domains we ran a grid search for each algorithm on the most important hyperparameters and selected the configuration that yielded the highest QD Score.

- For CMA-MEGA, we searched over initial step size of the ES ($\sigma_0$) and the optimizer learning rate.
- For CMA-MAEGA we searched over initial step size of the ES ($\sigma_0$), optimizer learning rate, and archive learning rate.
- For Sep-CMA-MAE we searched over initial step size of the ES ($\sigma_0$) and archive learning rate.
- For GA-ME we tuned iso and line sigma parameters and the gradient step size.

For LSI, we used the default hyperparameters from Fontaine & Nikolaidis (2023) used in the pyribs (Tjanaka et al., 2023b) implementation, with the only difference being the batch size, where we use 16 instead of 32 due to computational constraints. Furthermore, for DNS, we ran a similar grid search over the iso and line sigma parameters, the number of nearest neighbors ($k$), as well as the learning rate (for the DNS-G variant) in the IC domain to determine appropriate hyperparameters. We also chose the number of iterations such that all algorithms use (roughly) the same number of solution evaluations.

### D.3.2    SQUAD

By default, SQUAD uses the values presented in Table 7 for its hyperparameters and uses Adam (Kingma & Ba, 2015) to optimize its objective. Below, we will discuss the exceptions to these default values.

1. **LSI domain:** Due to computational constraints, we use a population size of 256 and a batch size of 8. We use $\gamma^2 = 0.01$ for the base version of the task and $\gamma^2 = 0.1$ for the hard version. We also use a larger learning rate of $0.1$ for the Adam optimizer.

2. **LP domain:** We use $\gamma^2 = 0.1$ for the easy version, $\gamma^2 = 0.5$ for the medium version, and $\gamma^2 = 1.0$ for the hard version.

Table 7: Default SQUAD parameters

| Parameter | Value |
|---|---|
| Population Size ($N$) | 1024 |
| Batch Size ($M$) | 64 |
| No. Neighbors ($K$) | 16 |
| Learning Rate | 0.05 |

We train SQUAD for 1000 iterations in LP and IC and for 175 iterations in LSI. The number of training iterations of baselines were set such that they use at least as many evaluations as SQUAD in all domains.

# E ADDITIONAL EXPERIMENTAL RESULTS

Here we provide more fine-grained statistics from the main experiments in the paper. Table 8 and 9 report the mean and max objectives, Vendi Score, and Coverage statistics of each algorithm in the LP and LSI domains, respectively. Table 10 reports the QD Score and QVS from the IC experiments. As noted in the paper, LP and IC results are averaged over 10 seeds and LSI results are averaged over 5 seeds.

We also provide hand-picked samples of the solutions found by SQUAD as well as the two best baselines, CMA-MEGA and CMA-MAEGA, in both LSI tasks (Figure 6 and Figure 7) and the IC domain (Figure 8). Lastly, Figure 9 compares the solutions found by SQUAD and CMA-M(A)EGA in the LSI domain when they are put in a traditional CVT archive. Since this domain has a 2-d behavior space, we can provide CVT archive visualizations for it.

Table 8: Additional statistics from LP experiments.

| Algorithm | Mean Objective | Max Objective | Vendi Score | Coverage |
|---|---|---|---|---|
| *easy (d = 4)* | | | | |
| SQUAD | $68.36 \pm 0.02$ | $89.28 \pm 0.15$ | $6.55 \pm 0.01$ | $86.4 \pm 0.3$ |
| CMA-MAEGA | $66.02 \pm 0.28$ | $91.00 \pm 0.43$ | $6.93 \pm 0.05$ | $98.7 \pm 0.2$ |
| CMA-MEGA | $66.40 \pm 0.26$ | $94.54 \pm 0.38$ | $7.61 \pm 0.10$ | $99.5 \pm 0.2$ |
| DNS | $68.07 \pm 0.06$ | $78.06 \pm 0.04$ | $1.63 \pm 0.00$ | $8.1 \pm 0.2$ |
| DNS-G | $78.23 \pm 0.12$ | $92.51 \pm 0.09$ | $1.35 \pm 0.01$ | $4.0 \pm 0.1$ |
| Sep-CMA-MAE | $69.81 \pm 0.28$ | $78.71 \pm 0.29$ | $1.25 \pm 0.01$ | $1.8 \pm 0.0$ |
| GA-ME | $69.76 \pm 0.95$ | $79.42 \pm 0.21$ | $1.07 \pm 0.01$ | $0.9 \pm 0.0$ |
| *medium (d = 8)* | | | | |
| SQUAD | $69.41 \pm 0.02$ | $87.70 \pm 0.36$ | $9.17 \pm 0.02$ | $93.1 \pm 0.2$ |
| CMA-MAEGA | $62.12 \pm 0.20$ | $84.41 \pm 0.33$ | $9.27 \pm 0.05$ | $100.0 \pm 0.0$ |
| CMA-MEGA | $62.13 \pm 0.67$ | $86.76 \pm 0.70$ | $7.58 \pm 0.19$ | $99.9 \pm 0.1$ |
| DNS | $66.87 \pm 0.08$ | $77.61 \pm 0.07$ | $1.67 \pm 0.00$ | $13.4 \pm 0.2$ |
| DNS-G | $75.96 \pm 0.07$ | $91.41 \pm 0.22$ | $1.43 \pm 0.00$ | $7.5 \pm 0.2$ |
| Sep-CMA-MAE | $66.49 \pm 1.42$ | $77.04 \pm 0.13$ | $1.25 \pm 0.02$ | $1.1 \pm 0.15$ |
| GA-ME | $69.48 \pm 1.08$ | $78.99 \pm 0.16$ | $1.07 \pm 0.01$ | $0.6 \pm 0.0$ |
| *hard (d = 16)* | | | | |
| SQUAD | $72.86 \pm 0.01$ | $83.92 \pm 0.21$ | $6.61 \pm 0.01$ | $91.1 \pm 0.2$ |
| CMA-MAEGA | $64.76 \pm 2.04$ | $81.27 \pm 0.66$ | $4.59 \pm 0.73$ | $71.8 \pm 13.3$ |
| CMA-MEGA | $58.11 \pm 0.39$ | $76.19 \pm 0.38$ | $3.73 \pm 0.08$ | $99.8 \pm 0.1$ |
| DNS | $66.19 \pm 0.04$ | $77.13 \pm 0.07$ | $1.69 \pm 0.00$ | $60.5 \pm 0.8$ |
| DNS-G | $74.29 \pm 0.09$ | $90.54 \pm 0.15$ | $1.45 \pm 0.00$ | $56.2 \pm 0.7$ |
| Sep-CMA-MAE | $65.04 \pm 0.84$ | $78.87 \pm 0.11$ | $1.33 \pm 0.01$ | $4.7 \pm 0.6$ |
| GA-ME | $78.83 \pm 0.66$ | $89.76 \pm 0.17$ | $1.08 \pm 0.00$ | $3.8 \pm 0.5$ |

Table 9: Additional statistics from LSI experiments.

| Algorithm | Mean Objective | Max Objective | Vendi Score | Coverage |
|---|---|---|---|---|
| *LSI* | | | | |
| SQUAD | $79.56 \pm 0.45$ | $83.81 \pm 0.09$ | $2.22 \pm 0.03$ | $32.4 \pm 0.5$ |
| CMA-MAEGA | $77.18 \pm 3.96$ | $86.98 \pm 0.45$ | $1.57 \pm 0.07$ | $15.9 \pm 2.0$ |
| CMA-MEGA | $83.38 \pm 0.89$ | $87.46 \pm 0.06$ | $1.68 \pm 0.01$ | $19.7 \pm 0.2$ |
| DNS | $-221.24 \pm 28.01$ | $84.10 \pm 0.11$ | $1.39 \pm 0.00$ | $10.2 \pm 0.3$ |
| DNS-G | $-288.33 \pm 26.12$ | $85.22 \pm 0.05$ | $1.36 \pm 0.00$ | $9.3 \pm 0.0$ |
| Sep-CMA-MAE | $-476.62 \pm 241.08$ | $-139.05 \pm 135.77$ | $1.01 \pm 0.01$ | $0.4 \pm 0.1$ |
| GA-ME | $-558.40 \pm 68.50$ | $83.72 \pm 0.10$ | $1.25 \pm 0.01$ | $6.8 \pm 0.3$ |
| *LSI (hard)* | | | | |
| SQUAD | $82.51 \pm 0.01$ | $84.26 \pm 0.09$ | $1.83 \pm 0.00$ | $6.0 \pm 0.2$ |
| CMA-MAEGA | $81.55 \pm 1.25$ | $87.05 \pm 0.12$ | $1.22 \pm 0.02$ | $0.9 \pm 0.2$ |
| CMA-MEGA | $84.24 \pm 0.52$ | $85.98 \pm 0.2$ | $1.10 \pm 0.02$ | $0.6 \pm 0.1$ |
| DNS | $-222.68 \pm 12.24$ | $84.02 \pm 0.04$ | $1.38 \pm 0.00$ | $10.2 \pm 0.2$ |
| DNS-G | $-214.70 \pm 35.08$ | $85.17 \pm 0.03$ | $1.35 \pm 0.01$ | $9.1 \pm 0.2$ |
| Sep-CMA-MAE | $-37.6 \pm 94.16$ | $16.74 \pm 42.55$ | $1.00 \pm 0.00$ | $0.2 \pm 0.0$ |
| GA-ME | $-168.09 \pm 217.38$ | $83.46 \pm 0.14$ | $1.04 \pm 0.01$ | $0.2 \pm 0.0$ |

Table 10: Additional statistics from IC experiments.

| Algorithm | QD Score | QVS |
|---|---|---|
| SQUAD | $5086.2 \pm 54.7$ | $457.35 \pm 0.23$ |
| CMA-MAEGA | $4605.8 \pm 40.5$ | $294.07 \pm 1.96$ |
| CMA-MEGA | $3565.7 \pm 386.2$ | $246.89 \pm 17.7$ |
| DNS | $1128.4 \pm 15.7$ | $115.17 \pm 0.65$ |
| DNS-G | $1148.0 \pm 24.2$ | $124.72 \pm 0.33$ |
| Sep-CMA-MAE | $348.9 \pm 17.6$ | $94.735 \pm 0.8$ |
| GA-ME | $140.9 \pm 23.2$ | $83.43 \pm 2.97$ |

# F    STATEMENT ON GENERATIVE AI USAGE

Generative AI tools were used as an aid to improve clarity and style in the writing of this paper.

# G    RUNTIME ANALYSIS

Table 11 summarizes the wall clock runtimes for SQUAD and all baselines on the three domains we consider. The most influential factor in SQUAD's runtime is the cost of gradient computation. In the Rastrigin domains this cost is relatively small. Therefore, even in the hard setting where the behavior space has dimension 16, SQUAD completes all 1000 iterations in under one minute. In these domains the computational structure is simple and backpropagation is inexpensive, which leads to very fast overall runtime.

In contrast, the IC and LSI domains require gradients that must be backpropagated through significantly more complex computational pipelines. In IC, each gradient step involves differentiation through a differentiable renderer. In LSI, gradients pass through both StyleGAN and CLIP, which are large networks and therefore incur substantial computational overhead. As a result, the wall clock times for SQUAD in these two domains are noticeably higher.

It is important to emphasize that these higher runtimes do not indicate inefficiency of SQUAD. In fact, SQUAD converges very quickly to high-quality solutions. Figure 10 shows the training curves of SQUAD together with the final performance of all baselines in terms of QD Score and QVS in the IC domain. SQUAD surpasses all baselines in both metrics in fewer than 200 iterations. Nevertheless, we ran SQUAD for 1000 iterations primarily to ensure an equal evaluation budget

**Generated Samples**

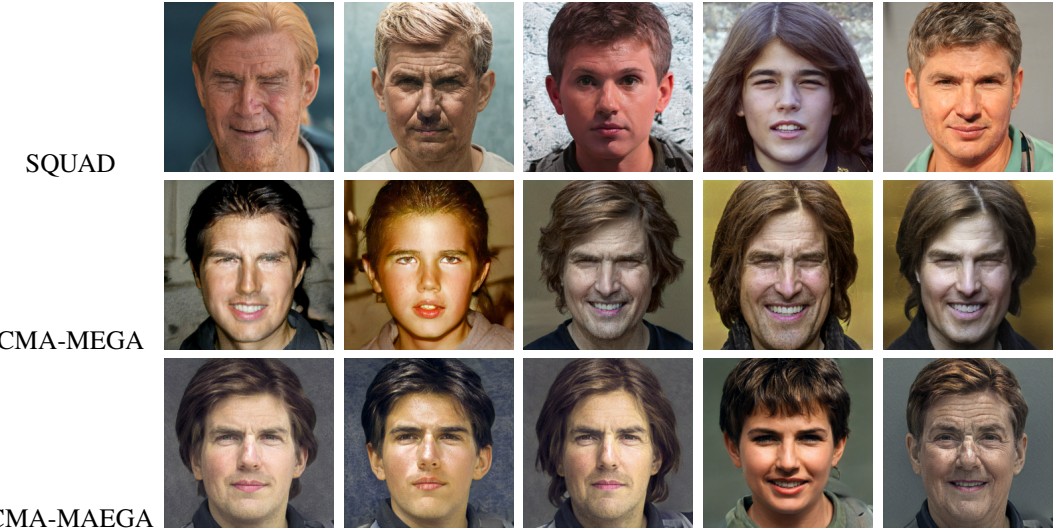

Figure 6: Qualitative comparison of SQUAD against two baselines in LSI. Each row corresponds to one algorithm, with five representative samples handpicked from the populations.

**Generated Samples**

Figure 7: Qualitative comparison of SQUAD against two baselines in LSI (hard). Each row corresponds to one algorithm, with five representative samples handpicked from the populations.

across algorithms. In practice, one can use SQUAD with a far smaller number of iterations and still obtain superior results, which directly reduces the wall clock time below the values reported in the table above.

**Generated Samples**

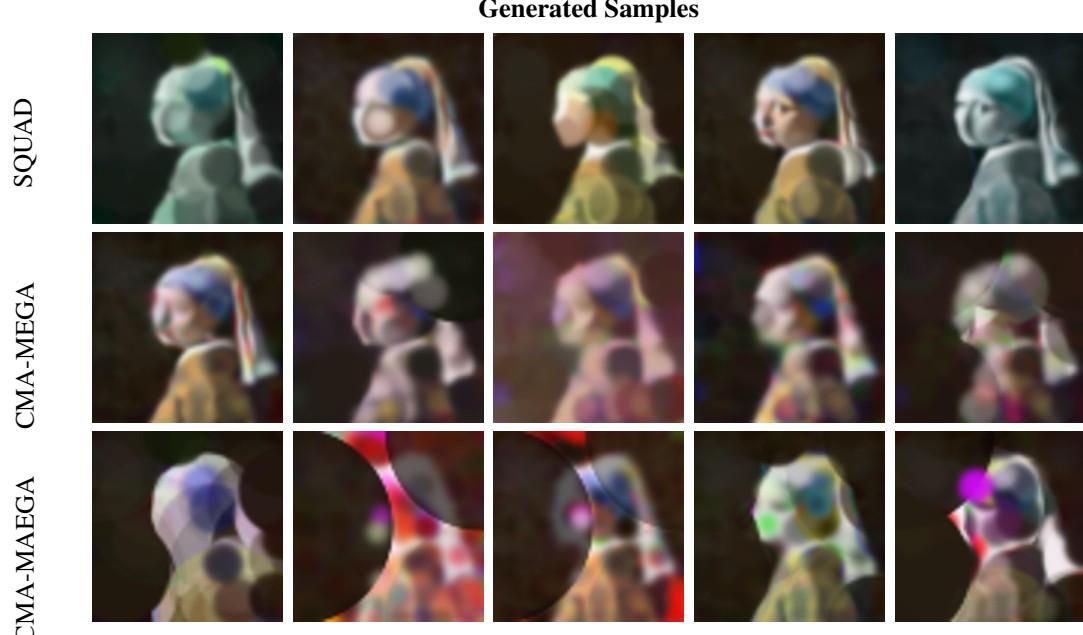

Figure 8: Qualitative comparison of SQUAD against two baselines in IC. Each row corresponds to one algorithm, with five representative samples handpicked from the populations.

Table 11: Wall clock runtime (in minutes) of SQUAD and all baselines across the three domains. For Rastrigin we report runtimes for the easy, medium, and hard settings. For LSI we report the base and hard settings.

| Method | Rastrigin | | | IC | LSI | |
|---|---|---|---|---|---|---|
| | Easy | Medium | Hard | | Base | Hard |
| SQUAD | <1 | <1 | <1 | 190 | 732 | 1316 |
| GA-ME | 1 | 3 | 17 | 66 | 403 | 707 |
| CMA-MAEGA | 45 | 78 | 66 | 8 | 222 | 339 |
| CMA-MEGA | 64 | 66 | 106 | 8 | 233 | 337 |
| DNS | 5 | 8 | 10 | 6 | 15 | 25 |
| DNS-G | 3 | 4 | 5 | 50 | 207 | 355 |
| Sep-CMA-MAE | 1 | 3 | 33 | 5 | 8 | 10 |

(a) **SQUAD**

(b) **CMA-MEGA**

(c) **CMA-MAEGA**

Figure 9: Final CVT archives of SQUAD, CMA-MEGA, and CMA-MEGA in the LSI domain.

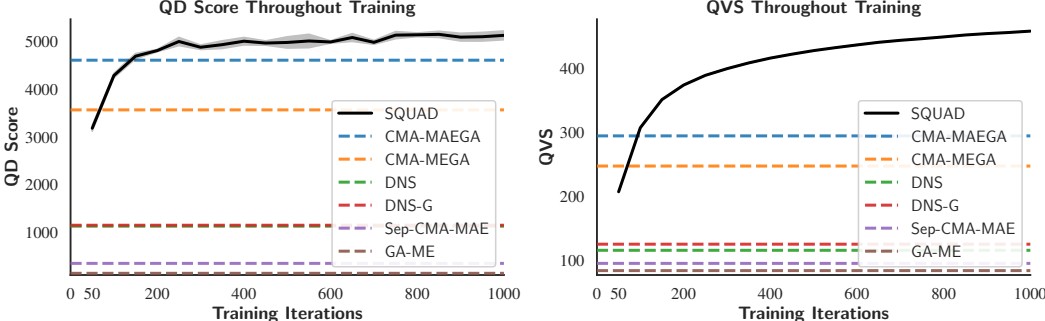

Figure 10: Training curves for SQUAD compared with the final QD Score and QVS values of all baselines in the IC domain. SQUAD exceeds all baselines on both metrics in fewer than 200 iterations.

