# OpenReview forum: "Soft Quality-Diversity Optimization"
_ICLR.cc/2026/Conference — ICLR 2026 Poster_

### Official Review · Reviewer_RoFt · 2025-10-19

**Soundness:** 2
**Presentation:** 2
**Contribution:** 2
**Rating:** 6
**Confidence:** 3

**Summary:**

This paper proposes a Soft Quality-Diversity (QD) optimization framework that avoids discretizing the behavior space. It defines the Soft QD Score, analyzes its properties (monotonicity, submodularity, limiting equivalence to traditional QD Score), and derives SQUAD, a differentiable algorithm based on a lower bound of the Soft QD Score. SQUAD optimizes via a quality term (maximizing solution quality) and a repulsive diversity term (spreading solutions). Experiments on Linear Projection, Image Composition, and Latent Space Illumination benchmarks show SQUAD outperforms baselines in high-dimensional scenarios.

**Strengths:**

The idea of "illumination" for addressing the high-dimensionality is interesting. A differentiable method of SQUAD is appreciated.

**Weaknesses:**

1. The core idea of continuous behavior space “illumination” overlaps with Kent et al. (2022)’s continuous QD Score. However, the paper only notes Kent et al.’s work as an evaluation tool, a more indepth discussion should be provided to support its novelty.
2. The paper claims SQUAD scales to high-dimensional spaces, but LP domain tests only go up to 16-dimensional behavior spaces. The impact of solution count  on computational efficiency and optimization performance is unexamined.
3. For baselines like CMA-MAEGA, grid search is used for LP/IC domains, but default parameters are adopted for LSI (due to computational constraints). Additionally, SQUAD uses a smaller population size (256 vs. default 1024) in LSI. This inconsistent tuning may prevent baselines from performing optimally.
4. QD often claims the diversity. In LSI, SQUAD outperforms baselines in QVS, but the paper does not validate if the “diversity” is semantically meaningful.

**Questions:**

1. what is the technical difference with Kent et al. (2022)’s continuous QD Score design?
2.  If generated "diverse" solutions differ in real-world attributes rather than just behavioral space coordinates?
3.  Why the logit transformation is selected for SQUAD, instead of other transformations  (e.g., Box-Cox, standardization)?

---

> ### Author Response · Authors · 2025-11-19
> **Part 1/2**
>
> We are thankful for your time and help in reviewing our paper. We are glad to hear that you found our work to be interesting and appreciable for the presentation of the SQUAD algorithm.
> We have revised the manuscript based on your suggestions. In particular, we added two new appendices where we perform an extra ablation of the solution count (population size) and discuss the runtimes in more detail. We have also included two additional baselines from the Novelty-search class of methods. Below, we will address your concerns and questions.
>
> > what is the technical difference with Kent et al. (2022)’s continuous QD Score design?
> > [...]  the paper only notes Kent et al.’s work as an evaluation tool, a more indepth discussion should be provided to support its novelty.
>
> We appreciate the reviewer's suggestion, and in the revised manuscript we expanded the related works section to more clearly articulate the differences between our SoftQD Score and the continuous QD Score of Kent et al. [1]. We summarize the key distinctions below.
>
> Kent et al. [1] introduce CQD Score as a metric for **evaluating** QD algorithms. While it shares with our SoftQD Score the idea of integrating over the behavior space, it differs in several important ways:
>
> CQD Score:
> 1. Must be estimated via Monte Carlo sampling.
> 2. Is not differentiable and cannot be used as an optimization objective.
> 3. Is used solely as an evaluation metric; no QD algorithm is proposed based on it.
> 4. Requires specifying distributions over both the behavior space and an additional "weight penalty space".
>
> In contrast, our work:
> 1. Theoretically motivates the proposed SoftQD Score and connects it to the well-established QD Score (Theorem 1).
> 2. Provides a tractable lower bound on SoftQD Score (Theorem 2).
> 3. Shows that this lower bound is fully differentiable and can be optimized directly with gradient-based methods.
> 4. Introduces SQUAD, a practical QD algorithm derived from this objective, and demonstrates its strong empirical performance.
> 5. Requires only a single intuitive parameter, $\gamma^2$, which directly controls the quality-diversity trade-off (Section 5.3).
>
> Overall, while our work and Kent et al. share the high-level "illumination" viewpoint (similar also to [2]), our contributions are fundamentally different: we formulate a differentiable objective for QD optimization and develop an effective algorithm that optimizes it, neither of which is present in [1].
>
> [1] Kent, Paul, et al. "A discretization-free metric for assessing quality diversity algorithms." _Proceedings of the Genetic and Evolutionary Computation Conference Companion_. 2022.
> [2] Cully, Antoine, et al. "Robots that can adapt like animals." _Nature_ 521.7553 (2015): 503-507.
>
> > The paper claims SQUAD scales to high-dimensional spaces, but LP domain tests only go up to 16-dimensional behavior spaces.
>
> We would like to clarify that the vast majority of existing QD work evaluates algorithms in very low-dimensional behavior spaces (typically 2-4D), largely due to the difficulties that QD algorithms face as dimensionality increases. For instance, the recent work of Bahlous-Boldi et al. [1] (which we included as a baseline following reviewer 8sBH's suggestion) also positions itself as addressing high-dimensional settings but reports experiments only up to 10D.
>
> In this context, our 16 and 8 dimensional behavior spaces in the Rastrigin domain, as well as the 7D behavior space in LSI-hard, already exceed the dimensionalities commonly used in the literature. This is reflected in the substantial degradation of most baselines in these settings. Demonstrating that SQUAD maintains strong performance in these challenging, higher-dimensional spaces suggests that it is more scalable than existing approaches.
> We hope that these results will help motivate the development of future QD benchmarks that explore even larger behavior spaces.
>
> [1] Bahlous-Boldi, Ryan, et al. "Dominated Novelty Search: Rethinking Local Competition in Quality-Diversity." _Proceedings of the Genetic and Evolutionary Computation Conference_. 2025.
>
> > The impact of solution count  on computational efficiency and optimization performance is unexamined.
>
> Thank you for this suggestion. In the revised manuscript, we added an additional experiment in Appendix C.4 that studies the effect of population size on SQUAD's performance. As expected, increasing the population size improves performance by allowing the algorithm to cover the behavior space more effectively. We also report the impact on runtime, where we observe a linear relationship between computational cost and population size.

---

> ### Author Response · Authors · 2025-11-19
> **Part 2/2**
>
> > For baselines like CMA-MAEGA, grid search is used for LP/IC domains, but default parameters are adopted for LSI (due to computational constraints). Additionally, SQUAD uses a smaller population size (256 vs. default 1024) in LSI. This inconsistent tuning may prevent baselines from performing optimally.
>
> Since IC is a newly introduced benchmark domain with no prior experimental results, we performed a grid search to fairly tune the baselines. We similarly tuned the baselines in the high-dimensional Rastrigin variants, as these high-dimensional settings had not been evaluated in previous work. In contrast, conducting a grid search in the LSI domain is prohibitively expensive due to its high computational cost. Fortunately, prior work [1, 2] provides recommended hyperparameters for this domain, and we used those values to ensure both feasibility and comparability.
>
> Regarding the smaller population size used by SQUAD in LSI (256 instead of 1024), this choice was also driven by computational constraints. However, as shown in our newly added ablations in Appendix C.4, SQUAD's performance consistently *improves* with larger populations. We therefore expect that using a larger population in the LSI domain would further strengthen SQUAD's results.
>
> [1] Fontaine, Matthew, and Stefanos Nikolaidis. "Differentiable quality diversity." _Advances in Neural Information Processing Systems_ 34 (2021): 10040-10052.
> [2] Fontaine, Matthew, and Stefanos Nikolaidis. "Covariance matrix adaptation map-annealing." _Proceedings of the genetic and evolutionary computation conference_. 2023.
>
> > If generated "diverse" solutions differ in real-world attributes rather than just behavioral space coordinates?
>
> > QD often claims the diversity. In LSI, SQUAD outperforms baselines in QVS, but the paper does not validate if the “diversity” is semantically meaningful.
>
> We agree that diversity in the behavior space should ideally correspond to meaningful semantic differences in the generated outputs. To assess this, we have provided qualitative comparisons between SQUAD and the strongest baselines in Appendix E, including visualizations from IC (Figure 8), LSI (Figure 6), and LSI-hard (Figure 7). Across these domains, we can see that SQUAD produces outputs that vary noticeably in "real-world attributes". For example, in the IC domain, SQUAD's solutions exhibit visible semantic variation such as distinct color palettes, brush-stroke smoothness, and stylistic differences, while maintaining high visual quality. In the more challenging LSI-hard domain, baseline methods tend to collapse to visually similar images, whereas SQUAD continues to produce a broad range of semantically distinct outputs.
>
> Thus, beyond the quantitative improvements, our qualitative results demonstrate that SQUAD's diversity is reflected in meaningful visual and semantic variation, rather than only in the behavior-space representation.
>
> > Why the logit transformation is selected for SQUAD, instead of other transformations (e.g., Box-Cox, standardization)?
>
> We would like to clarify that the logit transformation in SQUAD is used specifically to map a *bounded* behavior space $(0,1)^n$ into an *unbounded* space $\mathbb{R}^n$, which is required by the SoftQD objective derived in Section 3. The logit function is a natural choice because it is a smooth, strictly monotonic bijection from $(0, 1)$ to $\mathbb{R}$. Alternative transformations such as Box-Cox or standardization do not serve this purpose: Box-Cox assumes strictly positive *unbounded* inputs, while standardization merely rescales data within the same bounded support and does not remove the domain constraints.
> We agree that other bijective mappings from $(0,1)$ to $\mathbb{R}$ could potentially be explored. Nevertheless, the simple choice of the logit transformation allowed SQUAD to outperform the baselines in almost all domains. While it is possible that performance could be further improved with a different transformation, our experiments suggest that the logit transformation is already highly effective. We appreciate this suggestion and explicitly mention it as a design choice that future work could explore to further enhance performance in the revised manuscript.
>
>
> Thank you again for taking the time to review the paper and providing helpful feedback. We hope that these explanations and revisions sufficiently address your concerns. Please let us know if any further clarification or modification is required to satisfy your remaining concerns.

---

### Official Review · Reviewer_9uNt · 2025-11-01

**Soundness:** 3
**Presentation:** 4
**Contribution:** 3
**Rating:** 6
**Confidence:** 4

**Summary:**

This paper identifies a key limitation in current Quality-Diversity (QD) optimization methods: reliance on discrete archives (tessellations) to maintain diversity, which suffers from the curse of dimensionality in high-dimensional behavior spaces. To address this, the authors propose "Soft QD," a continuous relaxation of the QD problem. They define a new objective, the "Soft QD Score," which models each solution as a light source with a Gaussian field of influence, integrating the maximum influence over the entire behavior space.

Because directly maximizing this integral is intractable, the authors derive a differentiable lower bound that simplifies to a standard quality maximization term minus a pairwise repulsion term (weighted by quality and proximity). This leads to a new algorithm, SQUAD (Soft QD Using Approximated Diversity). Empirical results on Linear Projection (up to 16D behavior space), Image Composition, and Latent Space Illumination benchmarks demonstrate that SQUAD scales significantly better to high-dimensional behavior spaces than state-of-the-art differentiable QD baselines (like CMA-MAEGA) and offers a controllable trade-off between quality and diversity via its kernel bandwidth hyperparameter.

**Strengths:**

- Novel Problem Formulation: Shifting the QD objective from a discrete, archive-based metric to a continuous, integral-based "illumination" field is a significant and original theoretical contribution. It elegantly sidesteps the arbitrary nature of defining grid resolutions or pre-computing tessellations.

- Scalability to High-Dimensional Behavior Spaces: The empirical results strongly support the core claim that this method handles higher-dimensional behavior spaces better than archive-based counterparts. Figure 3 shows standard methods (CMA-MEGA/MAEGA) degrading at $d=16$ while SQUAD maintains performance. Besides, by framing the problem as a unified differentiable objective, SQUAD can seamlessly leverage modern optimizers like Adam, simplifying the typical QD loop.

- Theoretical Grounding: The derivation of SQUAD from the Soft QD Score via Bonferroni inequalities and geometric mean approximations (Theorem 2 and Appendix A) provides a solid theoretical foundation for the proposed pairwise repulsion objective, rather than just proposing it as a heuristic.

- Clarity and Presentation: The paper is exceptionally well-written. The conceptual difference between hard and soft archives is intuitively visualized in Figure 1.

**Weaknesses:**

- Hyperparameter Sensitivity ($\gamma$): The method's performance and its trade-off between quality and diversity are heavily dependent on the kernel bandwidth $\gamma^2$ (Section 5.3). While this provides controllability, it also introduces a critical hyperparameter that might be difficult to tune a priori for new domains compared to setting a grid resolution.

- Bounded Space Reliance: Appendix C.3 highlights that the logit transformation for bounded spaces is "critical" for success. This suggests the method might struggle with intrinsically unbounded behavior spaces or spaces where appropriate transformations are unknown, limiting generality slightly compared to robust binning strategies.

- Computational complexity of pairwise interactions: While standard QD archive insertions are typically $O(1)$ (grids) or roughly logarithmic (CVT trees) per solution, SQUAD's objective involves pairwise interactions. Although mitigated by KNN ($O(Nk)$), this could potentially become a bottleneck for very large populations compared to purely archive-based approaches, though it is likely an acceptable trade-off for high-dimensional capabilities.

**Questions:**

1. Adaptive Bandwidth: Given the strong influence of $\gamma^2$ on the quality-diversity trade-off (Figure 4), did you explore any mechanisms for adapting $\gamma^2$ online? For instance, starting with a larger $\gamma$ for broad exploration and annealing it for fine-grained illumination later in training?

2. Wall-clock time: Could you provide more details on the wall-clock time comparison between SQUAD (using its batch/KNN approximations) and the highly optimized pyribs baselines, particularly as the population size $N$ scales up?

3. Deceptive Landscapes: How does SQUAD behave in highly deceptive behavior landscapes? Does the continuous repulsion field ever prevent it from traversing narrow "corridors" in behavior space that a discrete archive might serendipitously cross?

---

> ### Author Response · Authors · 2025-11-19
> **Part 1/2**
>
> We are thankful for your time and help in reviewing our paper. We are glad to hear that you found our work to be novel, theoretically sound, and exceptionally well-written.
> We have revised the manuscript based on your suggestions. In particular, we added two new appendices where we discuss the runtimes and perform an extra ablation. We have also included two additional baselines from the Novelty-search class of methods. Below, we will address your concerns and questions.
>
> > Hyperparameter Sensitivity ($\gamma$): [...]
>
> We agree that $\gamma^2$ plays a critical role in SQUAD. While it may need to be tuned for a new domain to achieve desired performance, it also simplifies the design of the algorithm by providing a single, intuitive handle on the quality-diversity trade-off. As shown in Section 5.3, increasing $\gamma^2$ consistently improves diversity, while decreasing it consistently improves quality. This predictable behavior allows users to directly adjust the balance between quality and diversity. For example, if a user is not satisfied with the diversity of the solutions, they can simply increase $\gamma^2$, and when higher quality solutions are needed, they can decrease it. As such, $\gamma^2$ offers a simple and interpretable mechanism for tuning the trade-off, providing a flexibility that is not easily achieved with prior QD methods.
>
>
> > Bounded Space Reliance: [...]
>
> We would like to clarify that SQUAD's original objective assumes an *unbounded* behavior space and the logit transformation is only applied to handle *bounded* behavior spaces. You are correct that alternative transformations could affect performance. Nevertheless, the simple choice of the logit transformation allowed SQUAD to outperform the baselines in almost all domains. While it is possible that performance could be further improved with a different transformation, our experiments suggest that the logit transformation is already highly effective. We appreciate this suggestion and explicitly mention it as a design choice that future work could explore to further enhance performance in the revised manuscript.
>
> > Computational complexity of pairwise interactions: [...]
>
> You are correct about the $\mathcal{O}(Nk)$ cost of handling pairwise interactions. In our experiments, the population size $N$ is at most 1024, and in most practical applications we expect populations on the order of thousands. With the parallelization enabled by our JAX-based implementation, this should not be a bottleneck for typical use cases. Moreover, our ablation in Appendix C.2 shows that SQUAD is relatively robust to the choice of the number of neighbors, $k$. In extreme scenarios with millions of solutions and many neighbors, approximate $k$-nearest neighbor algorithms could be employed to mitigate computational costs.
>
>
> > Adaptive Bandwidth: [...]
>
> In preliminary experiments, we explored simple heuristics for adapting $\gamma^2$ based on the average distance between solutions, but found the results to be highly domain-specific: adaptive bandwidth improved performance in some domains and degraded it in others. We therefore opted for a fixed $\gamma^2$ for two reasons: (1) it is simple and already yields strong performance across domains, and (2) it provides an interpretable knob for trading off quality and diversity (Section 5.3). While adaptive bandwidths might further improve performance, they lack this interpretability. A systematic study of adaptive heuristics would be a promising direction for future work, but beyond the scope of the current paper, as noted in the conclusion.

---

> ### Author Response · Authors · 2025-11-19
> **Part 2/2**
>
> > Wall-clock time: [...]
>
> Thank you for this suggestion. We have added Appendix G, which reports the runtime of every algorithm in all domains. In brief, the dominant cost for SQUAD is the computation of gradients of objective and behavior descriptors. When these gradients are inexpensive to compute (e.g., Rastrigin domain) SQUAD is extremely fast, running in less than one minute even in 16-dimensional behavior spaces. In contrast, in domains such as LSI, gradient computation requires backpropagating through large pretrained models (StyleGAN and CLIP), and this results in significantly higher runtimes (up to ~20 hours in LSI-hard). In these settings, gradient-free baselines such as Sep-CMA-MAE and DNS run substantially faster, as expected.
>
> It is important to emphasize that our experimental design prioritized fairness in evaluation budget rather than runtime optimization. As a result, the reported SQUAD runtimes are conservative in that SQUAD can outperform the baselines before completing its full evaluation budget. For example, in the IC domain, SQUAD surpasses all baselines after fewer than 200 iterations, but we still run it for 1000 iterations to keep the same evaluation budget as the baselines. Appendix G includes a performance-vs-iterations plot illustrating this point.
>
> We highlight that SQUAD offers practical mechanisms to trade computation for speed when runtime is a bottleneck. For example, as shown in Appendix C.1, increasing the batch size reduces runtime without degrading performance, and users can also employ fewer training iterations while still achieving competitive results. Lastly, per your suggestion, we also analyzed the role of population size in SQUAD's performance. The results, shown in a new Appendix (C.4) indicate a clear trend of increased performance as a result of increased population size. The runtime of SQUAD had a linear relation with the population size, with the mean runtimes being 24, 48, 95, 190 minutes for population sizes of 128, 256, 512, and 1024, respectively (averaged over 3 seeds, with variances less than 10 seconds). Hence, reducing the population size can also help with the runtime, albeit at the cost of performance.
>
> > Deceptive Landscapes: [...]
>
> The repulsive term in SQUAD encourages solutions to spread out, but it does not create a global barrier that prevents exploration of low-density regions. If a “narrow corridor” in behavior space is accessible through a smooth gradient path (either via the fitness or BD gradients) then the repulsion from nearby solutions can actually help exploration by pushing some individuals toward less populated regions. However, SQUAD is ultimately a gradient-based method, and therefore it is not immune to local optima or discontinuities in the fitness-behavior landscape. In situations where the corridor is extremely narrow or requires traversing regions with vanishing or misleading gradients, a discrete-archive method might indeed discover it serendipitously, whereas SQUAD could struggle.
>
> In our benchmark suite, the closest analogue to this would be the Rastrigin domain, which combines a highly deceptive fitness landscape with a discontinuous behavior mapping (Figure 5 in Appendix D). While this does not fully match the reviewer’s definition of deceptive behavior landscapes, SQUAD’s strong performance there suggests some robustness to non-smooth behavior-fitness interactions.
> Ultimately, we see handling highly deceptive behavior landscapes as an important direction for future work, especially in future applications of SoftQD to RL domains, where such deceptive behaviors spaces emerge more naturally. In the revised version, we added a note about this to the conclusion section to highlight this avenue for further study.
>
>
> Thank you again for taking the time to review the paper and providing helpful feedback. We hope that these explanations and revisions sufficiently address your concerns. Please let us know if any further clarification or modification is required to satisfy your remaining concerns.

---

### Official Review · Reviewer_8sBH · 2025-11-07

**Soundness:** 3
**Presentation:** 4
**Contribution:** 3
**Rating:** 4
**Confidence:** 4

**Summary:**

Quality-Diversity (QD) algorithms typically discretize the behavior space into cells, find the best solution in each cell, and use the QD Score (i.e., the sum of the fitness value of the solution in each cell) as the objective. However, the discretizing mechanism makes it difficult to optimize end-to-end with gradient-based optimizers, and the number of cells grows exponentially when the dimension of the behavior space is large. For QD problems with differentiable fitness and behavior functions, this paper introduces a new objective, the Soft QD Score, to solve these issues. The behavior value is defined as the maximum fitness exponentially decreasing with distance among the population, and the Soft QD Score is defined as the integral of the behavior value over the behavior space. The Soft QD Score is then approximated by a computable and differentiable form, consisting of a quality term and a distance-based diversity term, which can be optimized directly with gradient-based optimizers. Experimental results on three differentiable QD (DQD) problems demonstrate that the proposed method outperforms other DQD methods significantly when the dimensions of behavior spaces are large.

**Strengths:**

- The proposed new objective bypasses the issues arising from behavior space discretizing, and enables it to be optimized by gradient-based optimizers easily.
- The proposed method performs well on DQD tasks with high-dimensional behavior spaces.
- The paper is well-organized and easy to follow. The code is available, contributing to reproducibility.

**Weaknesses:**

- The experiments are limited. The proposed method is not evaluated on high-dimensional reinforcement learning tasks, which are a more important type of DQD problem. Consequently, it is not compared with state-of-the-art methods that utilize policy gradients effectively.
- There are also other QD methods that do not discretize the behavior space (e.g., novelty search-based methods). They are not compared in the experiments.
- It is unclear whether the outstanding performance comes from the definition of the Soft QD Score, the end-to-end gradient-based optimization, or the population management.

**Questions:**

- How will the baselines perform if they use the Soft QD Score as the objective?
- What hyperparameter values do the baselines use in the experiments? These values may affect the performance of the baselines.
- Note that neighbor computing of batches might be time-consuming, since it applies $O(MNk)$ pairwise repulsions for each batch. Can you compare the running time of the algorithms?

---

> ### Author Response · Authors · 2025-11-19
> **Part 1/2**
>
> We are thankful for your time and help in reviewing our paper. We are glad to hear that you found our work to be novel and easy to follow.
> We have revised the manuscript based on your suggestions. In particular, we added two baselines from the Novelty-search class of methods as well as two new appendices where we discuss the runtimes and perform an extra ablation. Below, we will address your concerns and questions.
>
> > [...] The proposed method is not evaluated on high-dimensional reinforcement learning tasks [...]
>
> Strictly speaking, RL environments are not differentiable QD settings because the objective (expected return) and behavior descriptor functions of a policy are not differentiable. While gradient _estimates_ can sometimes be obtained through RL techniques, integrating such estimators into DQD algorithms is nontrivial.
> For example, the original differentiable QD work [1] introduced CMA-MEGA, which was not compatible with RL domains; extending it to RL environments required subsequent contributions such as DQD-RL [2] and PPGA [3]. Similarly, incorporating RL gradient estimators into SQUAD would require additional algorithmic development and careful experimentation. We view this as promising future work that is beyond the scope of the present paper, and we highlight it as such in the conclusion section.
>
> [1] Fontaine, Matthew, and Stefanos Nikolaidis. "Differentiable quality diversity." _Advances in Neural Information Processing Systems_ 34 (2021): 10040-10052.
> [2] Tjanaka, Bryon, et al. "Approximating gradients for differentiable quality diversity in reinforcement learning." _Proceedings of the Genetic and Evolutionary Computation Conference_. 2022.
> [3] Batra, Sumeet, et al. "Proximal Policy Gradient Arborescence for Quality Diversity Reinforcement Learning." _The Twelfth International Conference on Learning Representations_.
>
> > There are also other QD methods that do not discretize the behavior space (e.g., novelty search-based methods). They are not compared in the experiments.
>
> Your criticism is valid. In the revised version of the manuscript, we have added Dominated Novelty Search (DNS) [1] as a state-of-the-art representative of novelty-search-based QD methods. In addition, we included an enhanced variant, DNS-G, which incorporates gradient-based mutations (a la PGA-ME [2]) to complement the default evolutionary mutations used in DNS. The results show that both DNS and DNS-G outperform Sep-CMA-MAE and GA-ME in most settings, but they still fall short of CMA-M(A)EGA and our proposed method, SQUAD.
>
> [1] Bahlous-Boldi, Ryan, et al. "Dominated Novelty Search: Rethinking Local Competition in Quality-Diversity." _Proceedings of the Genetic and Evolutionary Computation Conference_. 2025.
> [2] Nilsson, Olle, and Antoine Cully. "Policy gradient assisted map-elites." _Proceedings of the Genetic and Evolutionary Computation Conference_. 2021.
>
> > It is unclear whether the outstanding performance comes from the definition of the Soft QD Score, the end-to-end gradient-based optimization, or the population management.
>
> We would like to clarify that the end-to-end gradient-based optimization and the elimination of discrete archives is a *direct consequence* of our reformulation of QD via the Soft QD Score. These are not independent design choices that can be ablated; rather, they are intrinsic elements of the SoftQD formulation and distinguish it from discretization-based approaches.
> That said, you are correct that aspects of population management can be ablated. In the revised manuscript, we added an experiment in Appendix C.4 that studies the effect of population size on SQUAD's performance. As expected, increasing the population size improves performance by allowing the algorithm to cover the behavior space more effectively.
>
> > How will the baselines perform if they use the Soft QD Score as the objective?
>
> Optimizing the Soft QD Score with existing baselines is not straightforward. Traditional QD Score is defined through a discretization of the behavior space, and many QD algorithms are tightly coupled to this discretized formulation. For example, CMA-MAEGA relies on maintaining per-cell thresholds that gradually increase as new solutions populate a given cell, while Sep-CMA-MAE ranks candidate solutions based on whether they fall into empty versus full archive cells.
>
> Because these mechanisms fundamentally depend on the structure of a discrete archive, it is not clear how such algorithms could be meaningfully adapted to optimize a discretization-free objective like the Soft QD Score. Developing such adaptations would require substantial algorithmic modifications that go beyond the scope of the present work.

---

> ### Author Response · Authors · 2025-11-19
> **Part 2/2**
>
> > What hyperparameter values do the baselines use in the experiments? [...]
>
> As described in Appendix D.3, we performed a grid search to tune the key hyperparameters of the baselines in the IC and Rastrigin domains. For the LSI domain, whose computational cost is substantially higher, we used the hyperparameters reported in prior work. Given the large number of methods and settings (7 algorithms across 6 domains), Appendix D.3.1 summarizes the most important shared hyperparameters, and details the specific hyperparameters over which we conducted the grid search for each algorithm.
>
> All hyperparameters used in our experiments are included in the submission's accompanying code. The `configs/` directory contains configuration files specifying the parameters of each algorithm separately, and the `scripts/` directory provides scripts for running the experiments in each domain. We consider the reproducibility of our work essential, as it allows other researchers to verify our findings and build on our framework of SoftQD. As such, we will open source the full code upon acceptance.
>
> > Note that neighbor computing of batches might be time-consuming [...]
>
> You are correct in that handling pairwise interactions incurs an $\mathcal{O}(Nk)$ for every solution update. In our experiments, the population size $N$ is at most 1024, and in most practical applications we expect populations on the order of thousands. With the parallelization enabled by our JAX-based implementation, this should not be a bottleneck for typical use cases. Moreover, our ablation in Appendix C.2 shows that SQUAD is relatively robust to the choice of the number of neighbors, $k$. In extreme scenarios with millions of solutions and many neighbors, approximate $k$-nearest neighbor algorithms could be employed to mitigate computational costs.
>
>  > [...] Can you compare the running time of the algorithms?
>
> Thank you for this suggestion. We have added Appendix G, which reports the runtime of every algorithm in all domains. In brief, the dominant cost for SQUAD is the computation of gradients of objective and behavior descriptors. When these gradients are inexpensive to compute (e.g., Rastrigin domain) SQUAD is extremely fast, running in less than one minute even in 16-dimensional behavior spaces. In contrast, in domains such as LSI, gradient computation requires backpropagating through large pretrained models (StyleGAN and CLIP), and this results in significantly higher runtimes (up to ~20 hours in LSI-hard). In these settings, gradient-free baselines such as Sep-CMA-MAE and DNS run substantially faster, as expected.
> It is important to emphasize that our experimental design prioritized fairness in evaluation budget rather than runtime optimization. As a result, the reported SQUAD runtimes are conservative in that SQUAD can outperform the baselines before completing its full evaluation budget. For example, in the IC domain, SQUAD surpasses all baselines after fewer than 200 iterations, but we still run it for 1000 iterations to keep the same evaluation budget as the baselines. Appendix G includes a performance-vs-iterations plot illustrating this point.
> Finally, we highlight that SQUAD offers practical mechanisms to trade computation for speed when runtime is a bottleneck. For example, as shown in Appendix C.1, increasing the batch size reduces runtime without degrading performance, and users can also employ fewer training iterations while still achieving competitive results (Appendix G).
>
>
> Thank you again for taking the time to review the paper and providing helpful feedback. We hope that these explanations and revisions sufficiently address your concerns. Please let us know if any further clarification or modification is required to satisfy your remaining concerns.

---

> ### Comment · Reviewer_8sBH · 2025-11-26
>
> Thank you for the detailed response. I have updated my score accordingly. I still have some concerns, though.
>
> > Optimizing the Soft QD Score with existing baselines [...]
>
> Optimizing the Soft QD Score with existing baselines would help clarify whether the performance gains primarily stem from the metric’s definition or from the end-to-end gradient-based optimization. Some baselines can be adapted straightforwardly. For example, you could replace the QD Score Improvement with the Soft QD Score Improvement in CMA-MEGA.
>
> > [...] runtimes [...]
>
> Thank you for providing the runtimes and training curves. As an additional suggestion, it would be great if you can also plot the training curves with runtimes on the x-axis, which might more directly support your claims.

---

> > ### Author Response · Authors · 2025-11-27
> >
> > We are glad to hear that our explanations addressed your main concerns. Below, we respond to the remaining points.
> >
> > **Training curves with runtime on the x-axis.**
> > Plotting training curves against runtime yields essentially the same curves as plotting them against iteration count. This is because each SQUAD iteration performs the same fully JAX-compiled computation, resulting in highly consistent runtimes of approximately 11.4 seconds per iteration, with very low variance (<0.5s). Consequently, replacing iterations with wall-clock time on the x-axis results in a nearly identical plot.
> >
> > **Replacing QD Score Improvement with Soft QD Score Improvement.**
> > As requested, we evaluated a modified version of CMA-MEGA that uses improvement in the approximate SoftQD Score (Equation 5), rather than in the QD Score, to rank solutions for the CMA-ES optimizer. The table below summarizes results on the IC domain (5 seeds). While the SoftQD-based variant shows modest improvements over the baseline CMA-MEGA in both QD Score and QVS, as well as exhibiting lower variance, the overall performance remains inferior to that of SQUAD.
> >
> > | Method                | QD Score         | QVS           | Mean Obj.    | Max Obj.     | Vendi Score | Coverage    |
> > | --------------------- | ---------------- | ------------- | ------------ | ------------ | ----------- | ----------- |
> > | **SQUAD**             | 5086.2 ± 54.7    | 457.35 ± 0.23 | 83.37 ± 0.02 | 93.58 ± 0.10 | 5.49 ± 0.00 | 5.68 ± 0.06 |
> > | **CMA-MEGA (SoftQD)** | 3750.51 ± 105.61 | 263.85 ± 3.32 | 73.51 ± 0.19 | 79.76 ± 0.48 | 3.59 ± 0.04 | 4.92 ± 0.14 |
> > | **CMA-MEGA (Base)**   | 3565.7 ± 386.2   | 246.89 ± 17.7 | 75.98 ± 0.26 | 86.18 ± 1.58 | 3.25 ± 0.24 | 4.54 ± 0.49 |
> >
> > As mentioned previously, adapting CMA-MEGA to SoftQD Score optimization is challenging because several of its design choices are closely tied to the structure of the discrete archive and the definition of QD Score. For example, in each iteration CMA-MEGA evaluates a batch of solutions and inserts them into the discrete archive using the standard QD replacement rule: if a new solution falls into a cell already occupied by another solution, it replaces the old one if it has higher fitness. This works seamlessly for QD Score, since each cell contributes independently and replacing a lower-performing solution always increases the score.
> > However, this logic does not transfer cleanly to SoftQD Score. Replacing a solution $x_{\text{old}}$ with a new one $x_{\text{new}}$ may *decrease* the SoftQD Score if $x_{\text{new}}$ lies closer to other archive solutions and therefore incurs a larger diversity penalty (Eq. 5). Unlike QD Score, SoftQD depends on the *relative arrangement* of all solutions, not just the best-in-cell.
> > One might consider a replacement rule that inserts $x_{\text{new}}$ only when it increases the SoftQD Score of the archive. However, this would fail to capture interactions among batch solutions: multiple new solutions might not be beneficial individually but could improve the SoftQD Score when added simultaneously. Accounting for such interactions requires evaluating all possible subsets of the batch (which is infeasible) or developing non-trivial approximation strategies.
> >
> > Such difficulties stem from the strong coupling between discrete archives, the definition of the QD Score, and algorithms such as CMA-MEGA that were specifically designed to optimize it. This tight integration makes straightforward adaptation to SoftQD optimization fundamentally challenging.
> >
> > Thank you once again for your thoughtful suggestions and stimulating questions. We hope the explanations above have resolved the remaining concerns, and we'd be happy to clarify anything further if needed.

---

### Author Response · Authors · 2025-11-30
**Summary of Rebuttal Discussions and Revisions**

Dear Area Chair,

As the discussion period has concluded, we are writing to summarize the productive exchange we had with the reviewers and the improvements we have made to the manuscript.

We were encouraged by the reviewers' positive reception. Reviewer 9uNt praised the **"novel problem formulation"** as a **"significant and original theoretical contribution,"** noting that it **"elegantly sidesteps the arbitrary nature of defining grid resolutions."** Reviewer 8sBH highlighted that our method effectively **"bypasses the issues arising from behavior space discretizing"** and outperforms other methods in high dimensions. Collectively, the reviewers commended the paper for being **"exceptionally well-written"** and **"easy to follow"** (Reviewers 8sBH, 9uNt). Crucially, following our detailed response and additional experiments, Reviewer 8sBH explicitly stated that we had addressed their concerns.

We provided comprehensive answers to the points raised by the reviewers. Below, we summarize **some** of the key outcomes of the discussion and the resulting improvements to the paper.

1. **Broader experimental validation & Ablation**
   To ensure a comprehensive evaluation, we addressed the suggestion to compare against Novelty Search-based approaches by adding Dominated Novelty Search (DNS) and DNS-G (gradient-enhanced) to the manuscript. SQUAD consistently outperforms both. Additionally, to address a follow-up question from Reviewer 8sBH regarding the source of our performance gains, we evaluated a modified variant of CMA-MEGA that optimizes the SoftQD Score. While SQUAD significantly outperformed this baseline too, it achieved some improvement over the baseline (unaltered) CMA-MEGA algorithm, confirming that both the SoftQD objective and our proposed end-to-end optimization procedure contribute to the enhanced performance.

2. **Further explanation about computational efficiency**
   Reviewers asked questions about the computational cost of our pairwise interaction computation ($\mathcal{O}(Nk)$). We expanded on the computational efficiency of SQUAD in our answers. Furthermore, following their suggestions, we added Appendix G, detailing wall-clock times for all algorithms. This analysis demonstrates that SQUAD's runtime is primarily driven by gradient computation for the objective and behavior descriptors. Consequently, in domains where gradients are cheap (e.g., Rastrigin), SQUAD is extremely fast (<1 min for 16D behavior spaces), while in complex domains, runtime is dominated by backpropagation through large pre-trained models (e.g., CLIP) rather than SQUAD's internal operations.

3. **Scaling the population size**
   We added Appendix C.4 to study the impact of population size, showing that SQUAD's performance improves consistently with larger populations. We also observed a linear relationship between runtime and population size, further validating our claim regarding the scalability of SQUAD.

4. **Clarifying the differences with prior work**
   In our response to Reviewer RoFt, we clarified the distinction between our work and Kent et al. (2022). Specifically, we noted that they proposed a non-differentiable *evaluation metric*, whereas our contribution is the derivation of a differentiable *optimization objective* (via a tractable lower bound). This innovation enables end-to-end gradient optimization and allows us to introduce a new QD algorithm, which the aforementioned work does not propose.

We believe SQUAD represents a significant step forward for Quality-Diversity optimization, offering the first fully differentiable training objective that effectively bypasses the limitations of discrete archives. We hope that our extensive rebuttal and the resulting improvements to the manuscript provide a strong basis for your decision.

Thank you for your time and consideration.
Sincerely,
The Authors

---

### Meta-Review · Area_Chair_4sh4 · 2026-01-08

**Summary:**

* Experimental Scope & Baselines: Multiple reviewers noted the absence of comparisons with Novelty Search-based methods and evaluated the algorithm only on differentiable tasks, omitting RL environments.

* Performance Attribution (Ablation): Reviewers questioned whether SQUAD's success was due to the Soft QD Score definition itself, the gradient-based optimization, or the population management strategies.

* Computational Efficiency: There were concerns regarding the $O(Nk)$ complexity of the pairwise interaction term, specifically how it scales with larger populations compared to traditional archive-based methods.

* Hyperparameter Sensitivity: The reliance on the kernel bandwidth $\gamma^{2}$ was seen as a potential critical hyperparameter that might be harder to tune than traditional grid resolutions.

* Novelty vs. Prior Work: Reviewer RoFt specifically questioned the technical distinction between this work and the continuous QD Score proposed by Kent et al. (2022).

**Reviewer Concerns:**

Addressed by Rebuttal

* Novelty Search Baselines: The authors successfully added Dominated Novelty Search (DNS) and its gradient variant (DNS-G) as baselines, showing SQUAD consistently outperforms them.

* Scaling and Population Size: Authors added Appendix C.4, demonstrating that SQUAD performance improves with larger populations and that the relationship between runtime and population size is linear.

* Comparison to Kent et al.: The authors clarified that Kent et al. provided a non-differentiable metric requiring Monte Carlo sampling, whereas SQUAD provides a differentiable optimization objective.

* Source of Performance Gains: Through a new ablation study using a modified CMA-MEGA that optimizes the SoftQD Score, the authors demonstrated that both the metric and the optimization procedure contribute to SQUAD's superiority.

* Semantic Diversity: Authors added qualitative visualizations in Appendix E, demonstrating that SQUAD's diversity translates to visible real-world stylistic and semantic variations.

Outstanding Concerns

* Reinforcement Learning Evaluation: The authors acknowledged that extending SQUAD to non-differentiable RL environments is nontrivial and remains future work.

* Adaptive Hyperparameters: While authors explored adaptive bandwidth ($\gamma^{2}$), they opted for a fixed value due to domain-specific results, leaving a systematic study of adaptive heuristics for the future.

* Deceptive Landscapes: The authors admit that in extremely narrow "corridors" of behavior space where gradients are misleading, discrete-archive methods might still be superior.

**Reviewer Scores:**

* 8sBH: Explicitly stated concerns were addressed and updated their score after the detailed response.


* 9uNt: Praised the "novel problem formulation" and "solid theoretical foundation". The wall-clock time data in Appendix G addressed their main secondary concern.

* RoFt: While the authors clarified the novelty vs. Kent et al., this reviewer had lower initial scores for Soundness and Presentation, which may only be partially mitigated by the new appendices.

---

### Decision · Program_Chairs · 2026-01-26

Accept (Poster)